# Comprehensive and unbiased multiparameter high-throughput screening by compaRe finds effective and subtle drug responses in AML models

Morteza Chalabi Hajkarim[1†], Ella Karjalainen[2†], Mikhail Osipovitch[1], Konstantinos Dimopoulos[3], Sandra L Gordon[1], Francesca Ambri[1], Kasper Dindler Rasmussen[4], Kirsten Grønbæk[1,3], Kristian Helin[1,5], Krister Wennerberg[1]*, Kyoung-Jae Won[1]*

[1]Biotech Research and Innovation Centre (BRIC) and Novo Nordisk Foundation Center for Stem Cell Biology (DanStem), University of Copenhagen, Copenhagen, Denmark; [2]Institute for Molecular Medicine Finland (FIMM), Helsinki Institute of Life Science, University of Helsinki, Helsinki, Finland; [3]Rigshospitalet, Copenhagen, Denmark; [4]Centre for Gene Regulation and Expression, School of Life Sciences, University of Dundee, Dundee, United Kingdom; [5]Cell Biology Program and Center for Epigenetics Research, Memorial Sloan Kettering Cancer Center (MSKCC), New York, United States

**\*For correspondence:**
krister.wennerberg@bric.ku.dk (KW);
kyoung.won@bric.ku.dk (K-JW)

[†]These authors contributed equally to this work

**Competing interest:** The authors declare that no competing interests exist.

**Abstract** Large-scale multiparameter screening has become increasingly feasible and straightforward to perform thanks to developments in technologies such as high-content microscopy and high-throughput flow cytometry. The automated toolkits for analyzing similarities and differences between large numbers of tested conditions have not kept pace with these technological developments. Thus, effective analysis of multiparameter screening datasets becomes a bottleneck and a limiting factor in unbiased interpretation of results. Here we introduce compaRe, a toolkit for large-scale multiparameter data analysis, which integrates quality control, data bias correction, and data visualization methods with a mass-aware gridding algorithm-based similarity analysis providing a much faster and more robust analyses than existing methods. Using mass and flow cytometry data from acute myeloid leukemia and myelodysplastic syndrome patients, we show that compaRe can reveal interpatient heterogeneity and recognizable phenotypic profiles. By applying compaRe to high-throughput flow cytometry drug response data in AML models, we robustly identified multiple types of both deep and subtle phenotypic response patterns, highlighting how this analysis could be used for therapeutic discoveries. In conclusion, compaRe is a toolkit that uniquely allows for automated, rapid, and precise comparisons of large-scale multiparameter datasets, including high-throughput screens.

## Editor's evaluation

This paper aims to address the current gap in the efficient analysis of large-scale multiparameter flow cytometry and other datasets. The authors offer a software toolkit with an efficient algorithm for comparing numerous samples at once. The study is well presented and is relevant to single cell analysis research.

**eLife digest** Biology has seen huge advances in technology in recent years. This has led to state-of-the-art techniques which can test hundreds of conditions simultaneously, such as how cancer cells respond to different drugs. In addition to this, each of the tens of thousands of cells studied can be screened for multiple variables, such as certain proteins or genes. This generates massive datasets with large numbers of parameters, which researchers can use to find similarities and differences between the tested conditions.

Analyzing these 'high-throughput' experiments, however, is no easy task, as the data is often contaminated with meaningless information, or 'background noise', as well as sources of bias, such as non-biological variations between experiments. As a result, most analysis methods can only probe one parameter at a time, or are unautomated and require manual interpretation of the data.

Here, Chalabi Hajkarim et al. have developed a new toolkit that can analyze multiparameter datasets faster and more robustly than current methods. The kit, which was named 'compaRe', combines a range of computational tools that automatically 'clean' the data of background noise or bias: the different conditions are then compared and any similarities are visually displayed using a graphical interface that is easy to explore.

Chalabi Hajkarim et al. used their new method to study data from patients with acute myeloid leukemia (AML) and myelodysplastic syndrome, two forms of cancer that disrupt the production of functional immune cells. The toolkit was able to identify subtle differences between the patients and categorize them into groups based on the proteins present on immune cells.

Chalabi Hajkarim et al. also applied compaRe to high-throughput data on cells from patients and mouse models with AML that had been treated with large numbers of specific drugs. This revealed that different cell types in the samples responded to the treatments in distinct ways.

These findings suggest that the toolkit created by Chalabi Hajkarim et al. can automatically, rapidly and precisely compare large multiparameter datasets collected using high-throughput screens. In the future, compaRe could be used to identify drugs that illicit a specific response, or to predict how newly developed treatments impact different cell types in the body.

## Introduction

Technological developments have accelerated the generation of large-scale multiparameter screening data through methodologies such as high-content microscopy and high-throughput flow cytometry (*Boutros et al., 2015*; *Saeys et al., 2016*; *Caraus et al., 2015*). These technologies can test hundreds of samples (such as drug treatments) each with tens of thousands of events (e.g. cells) labeled for numerous biomarkers (such as cytoplasmic or membrane markers). However, analyzing this massive multiparameter data to provide an overview of similarities and differences between hundreds of samples is still a challenge (*Boutros et al., 2015*; *Saeys et al., 2016*; *Caraus et al., 2015*). This analytical challenge is further complicated by various sources of bias and noise often existing in the data, such as batch effect and signal drift (a gradual shift in the marker intensity across a multi-well plate) (*Boutros et al., 2015*; *Saeys et al., 2016*; *Caraus et al., 2015*).

There have been efforts to cluster samples from large-scale multiparameter (multidimensional) screening data. A simple approach is to use a representative value for each cell marker such as median fluorescence intensity (MFI) for clustering samples (*Cossarizza et al., 2019*). However, using a single representative value can easily lead to loss of information about biologically relevant variance within and between cell subpopulations. Meta-clustering with single-cell clustering algorithms has been suggested to cluster samples based on the similarity of the centroids of cell subpopulations identified in the individual samples (*Qiu et al., 2011*; *Levine et al., 2015*; *Van Gassen et al., 2015*; *Ogishi et al., 2021*). While these algorithms are widely used in single-cell data analysis for clustering cells, they are not efficient for clustering of samples. This is because centroid-based analysis can be misleading when subclusters are not sufficiently distinct or the number of sub-clusters varies. Additionally, the heavy computing cost of meta-clustering makes it poorly suited for analyses of large datasets with many samples. Manual gating and machine learning based on prior knowledge have been used to cluster samples (*Amir et al., 2019*; *Bruggner et al., 2014*), but using prior knowledge for subpopulation identification can both lead to biased interpretations and failure to make de novo

discoveries. Dimension reduction methods (*Lvd and Hinton, 2008*; *Amir et al., 2013*; *McInnes et al., 2018*) coupled with the Jensen-Shannon divergence (JSD) metric have also been used to cluster multidimensional samples (*Amir et al., 2013*). These algorithms including factor analysis and principal component analysis (PCA) still require excessive computing costs with an inherent information loss. It is also important to note that none of the methodologies developed so far efficiently correct for sources of bias and noise in large-scale multiparameter screening data.

Available computational toolkits (*BioScience E, 2020*; *Potdar et al., 2020*; *Boutros et al., 2006*) mostly allow for single-parameter or unautomated analyses of large-scale screening data using the aforementioned methods. In these toolkits, each well should be first represented by a single parameter such as cell counts or centroids or they require manual intervention. To provide a useful toolkit for precise and effective interpretation of small- to large-scale multiparameter screening data, we developed compaRe. This toolkit has several unique modules for quality control, bias correction, pairwise comparisons, clustering, and data visualization. The quality control and bias correction modules can effectively reveal and remove various sources of bias in the screening data. compaRe clusters samples by measuring the similarity between them using a dynamic mass-aware gridding algorithm. This algorithm increases the robustness of the toolkit to the size of data and signal shift (a technical term referring to batch effect and signal drift), while guaranteeing fast clustering, as it does not bear the computing cost of dimension reduction and subsampling. The toolkit is available both as a

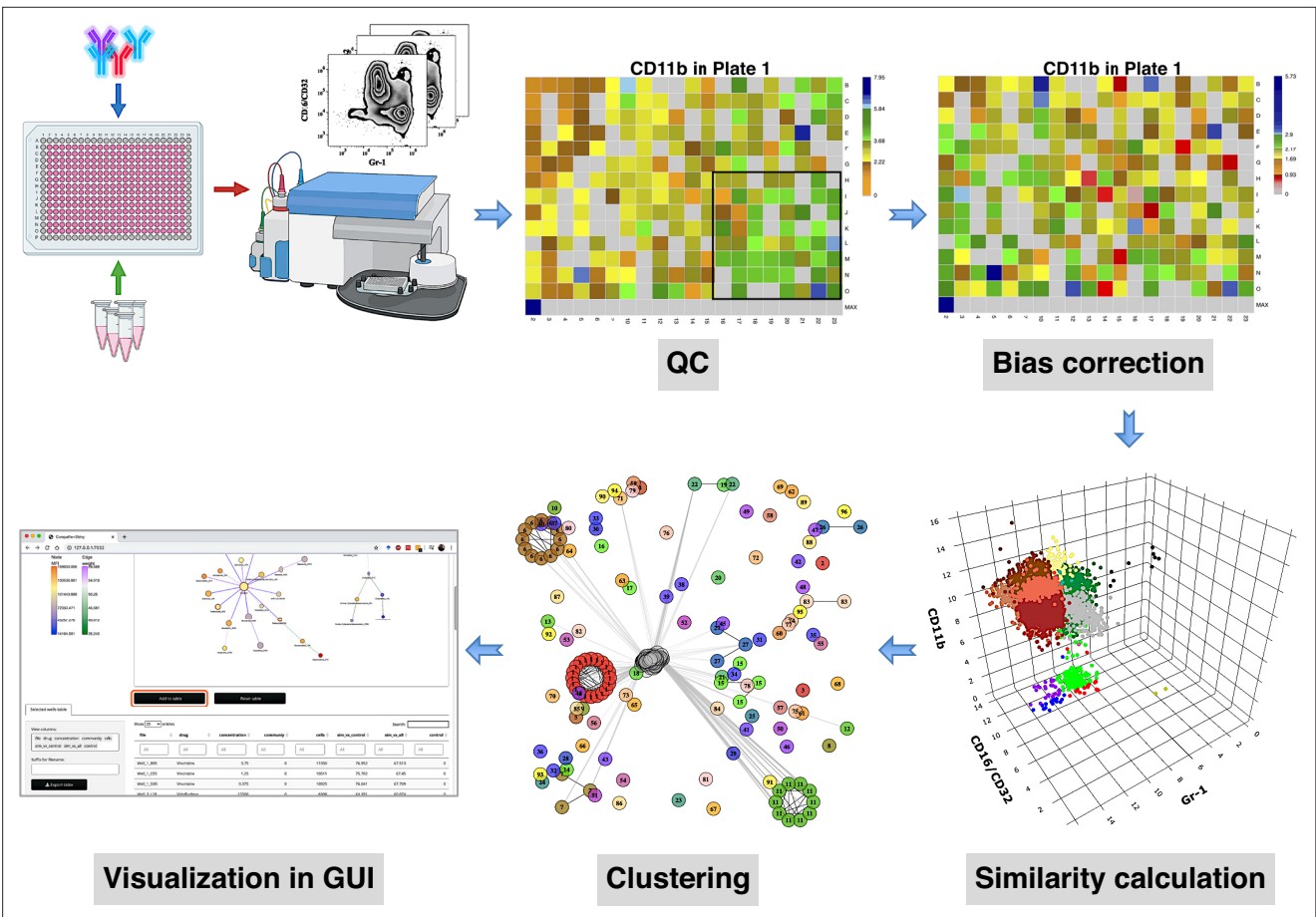

**Figure 1.** compaRe is a comprehensive suite for multiparameter screening data. High-throughput flow cytometry generates massive multidimensional data from hundreds of samples. compaRe's quality control (QC) module reveals several sources of bias in the assay such as signal (intensity difference between the top left and bottom right corners) and cell viability drifts. These two are corrected for in the bias correction modules within and between the plates. compaRe performs a pairwise similarity calculation between the samples using dynamic gridding and forming hypercubes (represented by distinct colors). The portions of the data within individual hypercubes are used to calculate similarity. Clustering is performed based on similarity. The graphical user interface (GUI) provides several ways to thoroughly explore and visualize the read-outs.

command-line version and a graphical user interface (GUI) version that provides various visualizations to help with the interpretation of its readouts.

compaRe performed robustly in the presence of background noise and batch effects even where these input data artifacts could not be corrected. compaRe analyses of multiparameter mass and flow cytometric data from acute myeloid leukemia (AML) and myelodysplastic syndrome (MDS) patient samples revealed interpatient heterogeneity and recognizable phenotypic profiles. When applied to high-throughput flow cytometry of the dose response of AML samples treated with various drugs, compaRe successfully corrected for various sources of bias and clustered the samples based on their response to treatment, allowing for detection of both drastic and subtle phenotypic responses.

## Results

### compaRe is a comprehensive toolkit for multiparameter screening data

compaRe is designed to analyze the data from small to large-scale multiparameter screening assays such as high-throughput flow cytometry, high-content microscopy, mass cytometry, and standard flow cytometry. The toolkit comprises several modules for quality control, bias correction, clustering, and visualization. *Figure 1* shows the modules for a high-throughput flow cytometry of AML samples taken from a mouse model treated with various drugs. During quality control, several sources of bias such as autofluorescence, bioluminescence, carryover effect, edge effect, signal drift, and cell viability drift (drift in the number of live cells across the plate) were identified. The bias correction module could effectively correct for signal and cell viability drifts (two main sources of bias in high-throughput screening with fluorescent markers) using regression analysis (*Figure 1*, Materials and methods).

At the core of the compaRe toolkit is a module for pairwise comparisons of samples. It measures the similarity between two samples using a dynamic mass-aware gridding algorithm (*Figure 1*, Materials and methods). Given two samples, the algorithm divides the higher dimensional space (formed by, for example, cell surface markers) of the samples individually into several spatial units called hypercubes. The average difference between proportions of data points present in corresponding hypercubes across the samples is used to represent similarity. In this setting, the module becomes robust to signal shift and data size difference between the two samples (Appendix 1). This module generates a similarity (affinity) matrix for the clustering module.

The clustering module uses a graphical algorithm (*Figure 1*, Materials and methods). Initially, all nodes (samples) are connected forming a complete weighted graph wherein weights represent similarity values. The graph is then pruned to remove potential false positive edges using a threshold inferred from negative controls (untreated samples). After constructing a linked graph, clustering is tantamount to finding maximal cliques (complete subgraphs that cannot be extended), each containing samples with similar responses. compaRe benefits from parallel computing and modular design. Its modular design allows the modules to run independently; thus, the similarity and clustering modules of compaRe can be potentially applied to any problem space.

### compaRe is ultra-fast and robust to background noise and batch effect

To evaluate the robustness of compaRe's comparison module to noise and batch effect, we benchmarked it against JSD with UMAP (for simplicity just JSD) and meta-clustering with PhenoGraph (for simplicity just meta-clustering) (*Levine et al., 2015*). We analyzed the publicly available mass cytometry data of a total of 21 bone marrow aspirate samples collected from 16 pediatric AML patients and five healthy adult donors labeled for detection of 16 cell surface markers (*Levine et al., 2015*). We introduced random noise with Gaussian distribution to the 16 parameters of each sample to simulate a batch effect. In this setting, although the added noise undermines similarity, the overall cell population configuration remains intact, and consequently the simulated samples will still have the highest similarity with their original samples.

Even with the added noise, the comparison module correctly identified similar samples (*Figure 2a*). Conversely, the batch effect seriously compromised the performance of both meta-clustering and JSD, showing several maximum similarities other than the originals (*Figure 2b and C*). In additional comparison with FlowSOM and SPADE, other commonly used tools for flow cytometry, compaRe's performance far exceeded their performance (*Appendix 1—figure 1*). This result demonstrates the

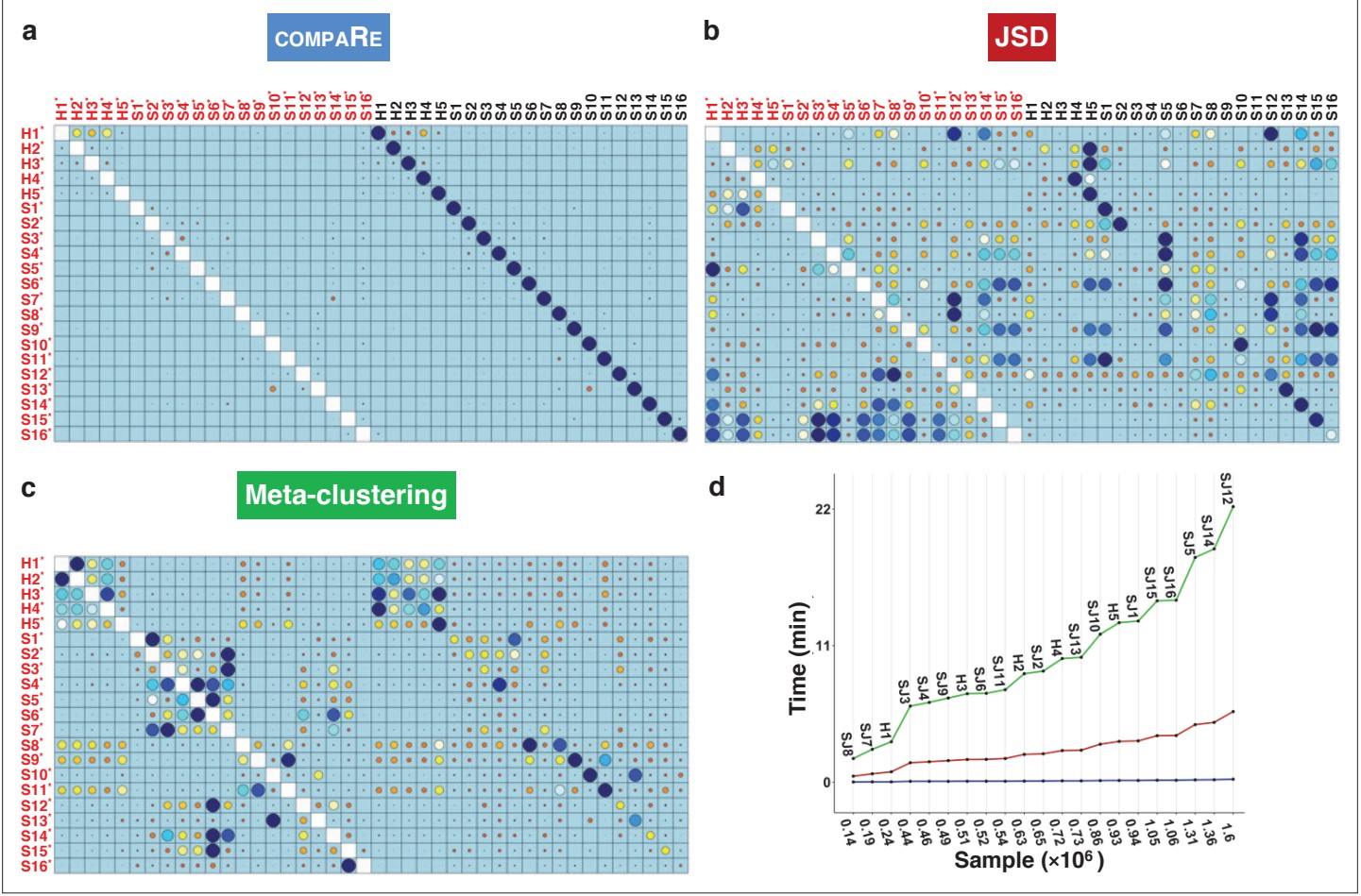

**Figure 2.** compaRe robustly measures the similarity between samples in the presence of batch effect. Similarity matrix generated by compaRe is shown in (**a**). Size and color of dots represent the level of similarity. Self-comparisons were removed. Noise was added (marked with *) to the original 21 mass cytometry samples of bone marrow aspirates from 16 pediatric AML patients (S) and five healthy adult donors (H). Similarity matrices using JSD with UMAP and meta-clustering with PhenoGraph are shown in (**b**) and (**c**), respectively. The run time of comparing each sample to itself is shown in (**d**). Samples were sorted based on their size.

advantage of using dynamic gridding for comparison of samples in the presence of noise or batch effect.

Notably, compaRe took only 25 min to analyze the 21 samples (210 pairwise comparisons), without subsampling or dimension reduction. Meanwhile, meta-clustering and JSD took 39 hr and 10 hr respectively. For the feasibility of JSD, we subsampled each sample to 100,000 cells (default value suggested in **Amir et al., 2013**). When we fixed this limit to 60% of each sample, the computing time of JSD increased to 3 days. To investigate the relation between run time and sample size, we compared each sample to itself and sorted measured times based on sample size (**Figure 2d**). The run time increased steeply for both meta-clustering and JSD as the sample size increased, while the increase for compaRe was almost unnoticeable.

To further show that compaRe can identify phenotypic changes from a high-dimensional dataset, we used a subset of the data with three healthy and two AML samples stained with 29 (15 membrane and 14 intracellular signaling) markers (**Appendix 1—figure 2**). Taking H1 as reference, we gradually removed 25%, 50%, 75%, and 100% of cells from a target cluster identified by PhenoGraph. The gradual removal can be regarded as a phenotypic change and the 75% reduction can potentially resemble a rare cell population (a small cluster of cells). As shown in the UMAP projections, the similarity decreased concurrently and more drastically after 100% reduction when phenotypic changes were detected, indicating compaRe is sensitive to phenotypic changes and the existence of rare cell populations.

## compaRe reveals interpatient similarity

Non-AML myeloid neoplasias such as MDS can evolve to become AML. Over time, about one-third of all MDS cases develop into AML (*DeVita and Lawrence, 2015*; *Niederhuber et al., 2020*). The risk of developing AML largely depends on the MDS subtype at the time of diagnosis, with high-risk MDS developing into AML more often than the lower-risk MDS subtypes (*Greenberg et al., 2012*). As many immunophenotypic abnormalities are not unique to MDS, several diagnostic flow cytometric antibody panels have been proposed (*van Dongen et al., 2012*; *Alhan et al., 2016*). The EuroFlow AML/MDS antibody panel (*van Dongen et al., 2012*) aims at the parallel identification and categorization of AML and MDS. Both diseases are heterogeneous, affecting multiple cell lineages and multiple maturation stages. Therefore, this panel concerns major myeloid lineages (neutrophilic, monocytic and erythroid) and the detection of abnormal lymphoid maturation profiles in four tubes. The panel uses four backbone markers to identify myeloblasts and an additional set of 15 markers devoted to the characterization of myeloid lineages (*Supplementary files 1 and 2*).

Unlike the backbone markers, the characterization markers are divided into each tube exclusively. This design was made so that characterization markers from different tubes can be inferred on the same backbone marker subpopulations, but the design makes it impossible to form a multiparameter dataset which is required for clustering methods. However, as compaRe's comparison module can compare cell population morphologies even in subspaces, we were able to use it to measure similarities between patient samples.

We analyzed 25 bone marrow mononuclear cell samples collected from 16 MDS patients and 9 AML patients (*Supplementary file 3*). The comparison module provided a detailed overview of similarities of samples. As expected, the AML samples exhibited a great amount of interpatient heterogeneity compared to the MDS samples (*Figure 3a and b*) with all MDS samples clustered together, and the AML samples spread over three clusters. To verify the performance of the module, we visualized the pairwise comparisons using UMAP projection (*Figure 3c* and *Appendix 1—figures 3–26*). The measured similarities perfectly matched the projections so that from top left to bottom right, as the similarity decreases, the degree of overlap decreases, and the number of exclusive cell populations increases.

We further investigated how different the three groups of the AML samples were (*Figure 4* and *Appendix 1—figure 27*). AML samples 1 and 9 of the blue cluster were confirmed to have a high degree of monocytic differentiation with marked expression of the monocytic maturation markers CD14, CD35, CD64, and CD300e. The AML samples of the green cluster, on the other hand, represented a cluster of poorly differentiated AML cases with low expression of differentiation markers and high expression of the stem cell/progenitor markers CD34 and CD117. Unlike the blue cluster with high monocytic differentiation, and the green cluster with poor monocytic differentiation, the AML samples 2 and 5 of the red cluster included both positive and negative populations of CD11b which is a common granulocytic and monocytic maturity marker, a feature observed in all MDS samples as well (*Appendix 1—figure 27*).

In conclusion, compaRe's comparison module can be used to optimize true cytometric n-dimensional immunophenotypic characterization of patient samples. Interpretation can then be performed in a conventional manner assisted by lower-dimensional projection tools such as PCA and UMAP that promptly provide a phenotypic profile of the patient samples.

## Identifying cell-subtype-specific drug responses in mouse AML cells

We applied compaRe to high-throughput flow cytometry data to identify cell subtype-specific responses evoked by antineoplastic agents in leukemic spleen cells from an AML mouse model. Splenic cells were sorted for c-Kit cell surface expression, allowing for the enrichment of stem/progenitor-type leukemic cells. On ex vivo expansion, these cells continuously expand and differentiate in a similar way as in vivo with a clear stem cell/progenitor population and partial differentiation towards CD11b/Gr-1 or CD16/CD32-expressing myeloid cells. After ex vivo expansion, the leukemic cells were plated onto multi-well plates containing a library of 116 antineoplastic agents including surface and nuclear receptor inhibitors and activators, enzyme inhibitors and, cytotoxic chemotherapy in a five-point concentration range, as well as 20 negative control wells (*Supplementary file 4*). After 72 hr of drug exposure, we stained the cells with fluorescently labeled antibodies against three cell surface markers (CD16/32, Gr-1 and CD11b) and quantified cell surface marker expression using a high-throughput flow cytometer.

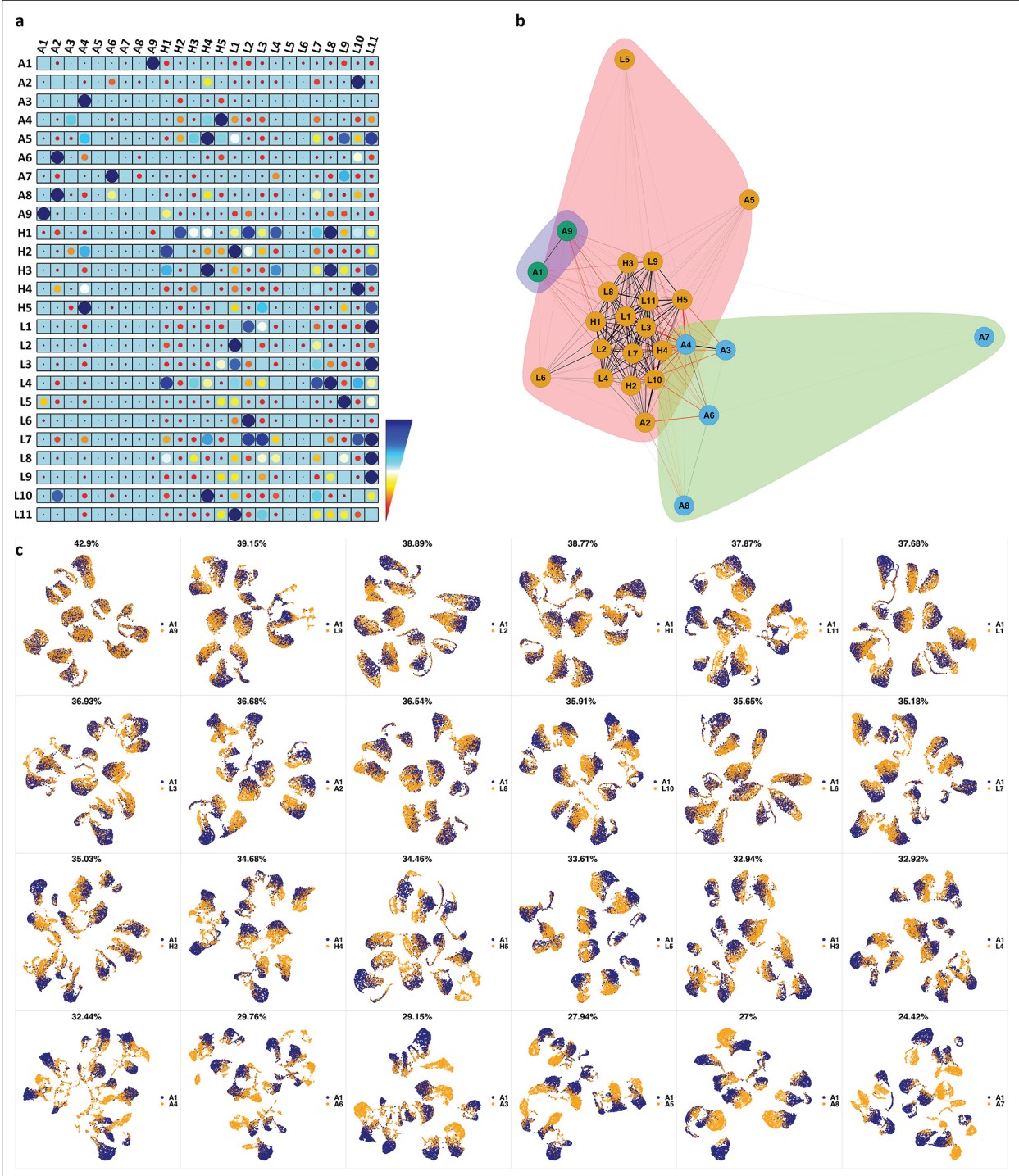

**Figure 3.** compaRe highlights immunophenotypic similarities. (**a**) The similarity band plot visualizes the similarity between a sample specified by its row (band) and other samples measured by compaRe (H: higher-risk MDS, L: lower-risk MDS and A: AML). Each band was independently transformed by an exponential function to emphasize the highest and the lowest similarity values. (**b**) A graphical representation of the similarities. The graph nodes (samples) were clustered by a random walk. (**c**) The UMAP projection of A1 sample against the other patient samples is provided as an example. The other projections are given in *Appendix 1—figures 3–26*. The projections were sorted based on similarity.

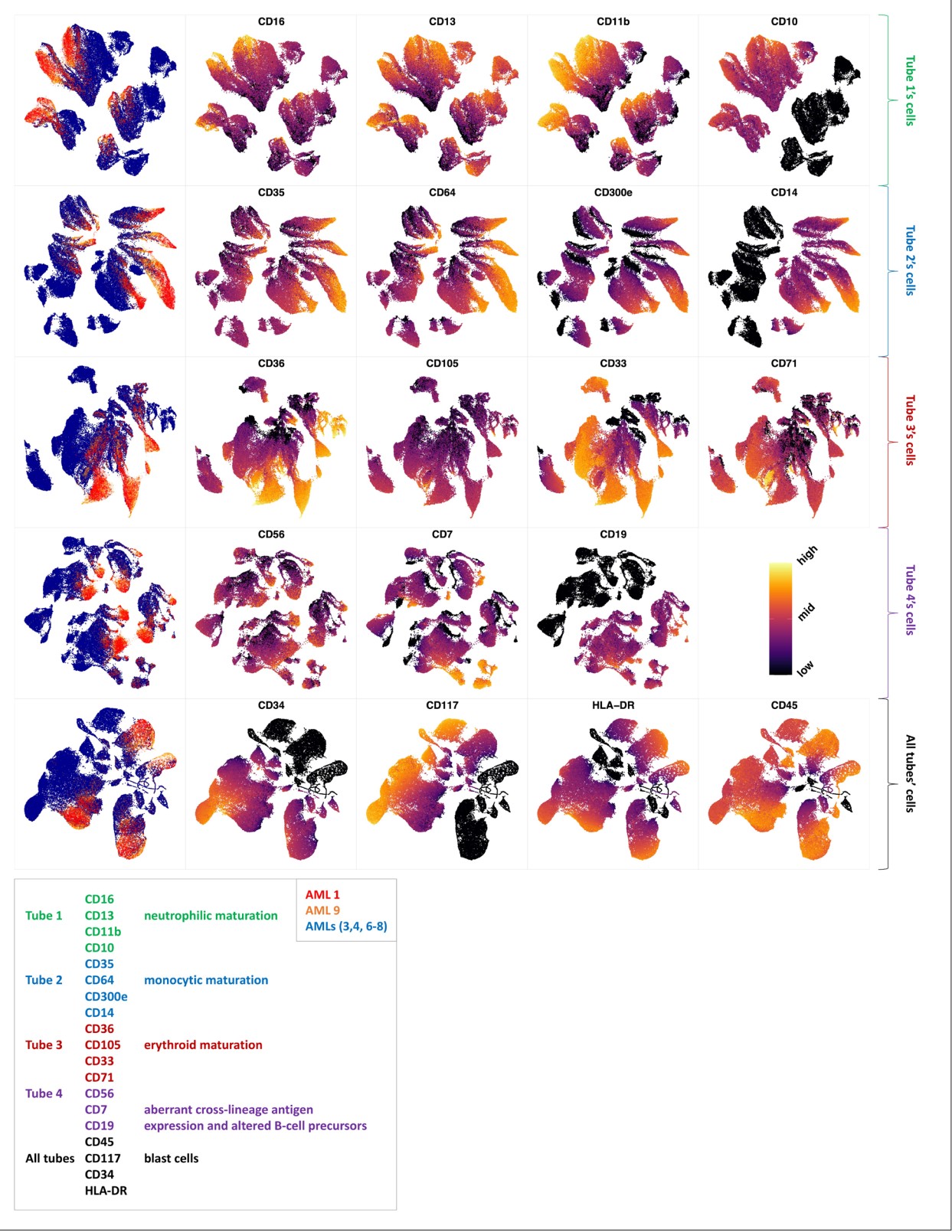

**Figure 4.** Immunophenotypic profiles of two different groups of AML patients. Each row shows the UMAP projection of AML samples 1 and 9 (red and orange) vs AML samples 3, 4, 6-8 (blue) of the green cluster of *Figure 3b* stained by the markers available in each tube.

compaRe corrected the intraplate signal drift, sources of bias in cell numbers, as well as interplate sources of bias (*Appendix 1—figure 28*). After clustering and clique analysis, we obtained 134 cliques, each sharing similar drug responses (*Supplementary file 5*).

To get an overview of the assay, we generated a dispersion map of the clusters (*Figure 5a and b* and Materials and methods). We identified a distinct response group characterized with decreased Gr-1 and concomitant increase of CD16/CD32 as compared to control (Group one in *Figure 5a*). Most of the cliques included in this response group consisted of drugs in high concentrations with cytotoxic/cytostatic effects. However, some drugs in this group had a milder effect on live cell numbers, and these were enriched for mitogen-activated protein kinase (MAPK) pathway-associated inhibitors (*Figure 5c*, *Supplementary file 6*). For instance, trametinib (2.5 nM) in clique 23 (C23) showed a marked decrease of Gr-1 and increase of CD16/CD32, further confirming the results of compaRe (*Figure 5d*). The MAPK pathway is a regulator of diverse cellular processes such as proliferation, survival, differentiation, and motility (*Dhillon et al., 2007*). Our findings suggest that MAPK signaling controls the differentiation and/or proliferation towards Gr-1-/CD16+ cells.

In high concentration, molibresib and birabresib, inhibitors of BET proteins BRD2, BRD3, and BRD4, caused a reduction in live cell counts but also a reduction of MFI in all the measured markers, which corresponds to the loss of differentiation marker positive cells (Gr-1+, CD11b+, CD16/CD32 high) (*Figure 5b*: C100, C110, *Figure 5d*). The BRD2/3/4 proteins regulate transcription via recognition of acetylated lysines on histones and concomitant recruitment of other transcription and chromatin remodeling factors to enhance transcriptional activity (*Ferri et al., 2016*). The enrichment of undifferentiated cells could therefore be due to an early block in differentiation or that inhibition of BRD2/3/4 has led to a general decrease of cell surface protein transcription.

In this cell model, the leukemic stem-like cells are expected to be present within the differentiation marker negative population. These cells are potential targets for treatments against leukemia. We observed response group 2 (*Figure 5a*) had a higher MFI in marker Gr-1 as compared to control, the increase was very slight and seemed to be linked to toxic drug concentrations. However, three drugs, vincristine (C80), tazemetostat, and tretinoin clearly reduced the proportion of differentiation marker negative cells (*Figure 5d*). Interestingly, these three drugs have distinct modes of action: vincristine is a microtubule polymerization inhibitor, tazemetostat inhibits the histone methyltransferase EZH2, and tretinoin is a retinoic acid receptor agonist (*Supplementary file 6*).

Taken together, compaRe analysis of the high-throughput flow cytometry screening data allowed rapid identification of several distinct phenotypic responses in this mouse AML model, as well as the cellular signals that drive them. Drugs of different mechanism of action can still cluster together if the cellular processes they affect converge in a specific model. Drug response in association with genetic alterations can be one of the applications of compaRe. The genetic alteration could be visualized in the clusters that compaRe identifies.

## Identifying highly selective signal transduction inhibitors in human AML cells

We further applied compaRe to the drug screening data from an AML patient sample. Primary AML bone marrow mononuclear cells were dispensed into a 384-multiwell plate containing a library of 40 drugs and drug combinations in seven-point concentration ranges (*Supplementary file 7*). After 72 hr of drug exposure, the cells were stained with fluorescently labeled antibodies against a panel of AML-related cell surface markers (CD45, CD34, CD38, CD117, HLA-DR, CD45-RA, CD3 and a mix of myeloid differentiation-related markers). A high-throughput flow cytometer was used to quantify cell surface marker expression.

compaRe analysis identified several distinct response groups (*Figure 6a*, *Supplementary file 8*). Response group one had notably higher MFIs in the CD34 and CD38 channels compared to controls. Interestingly, the increase in MFIs was due to a drug concentration-dependent appearance of a CD34+/CD38+ cell population that was barely detectable in the DMSO control samples (*Figure 6b*). The appearance of this CD34+/CD38+ population was also concomitant to a general increase in live cell count (*Figure 6c*). Altogether, seven different drugs had the same effect (*Figure 6d*), most of them being selective signal transduction inhibitors such as trametinib (MEK inhibitor), copanlisib (PI3K inhibitor), and PIM447 (PIM kinase inhibitor).

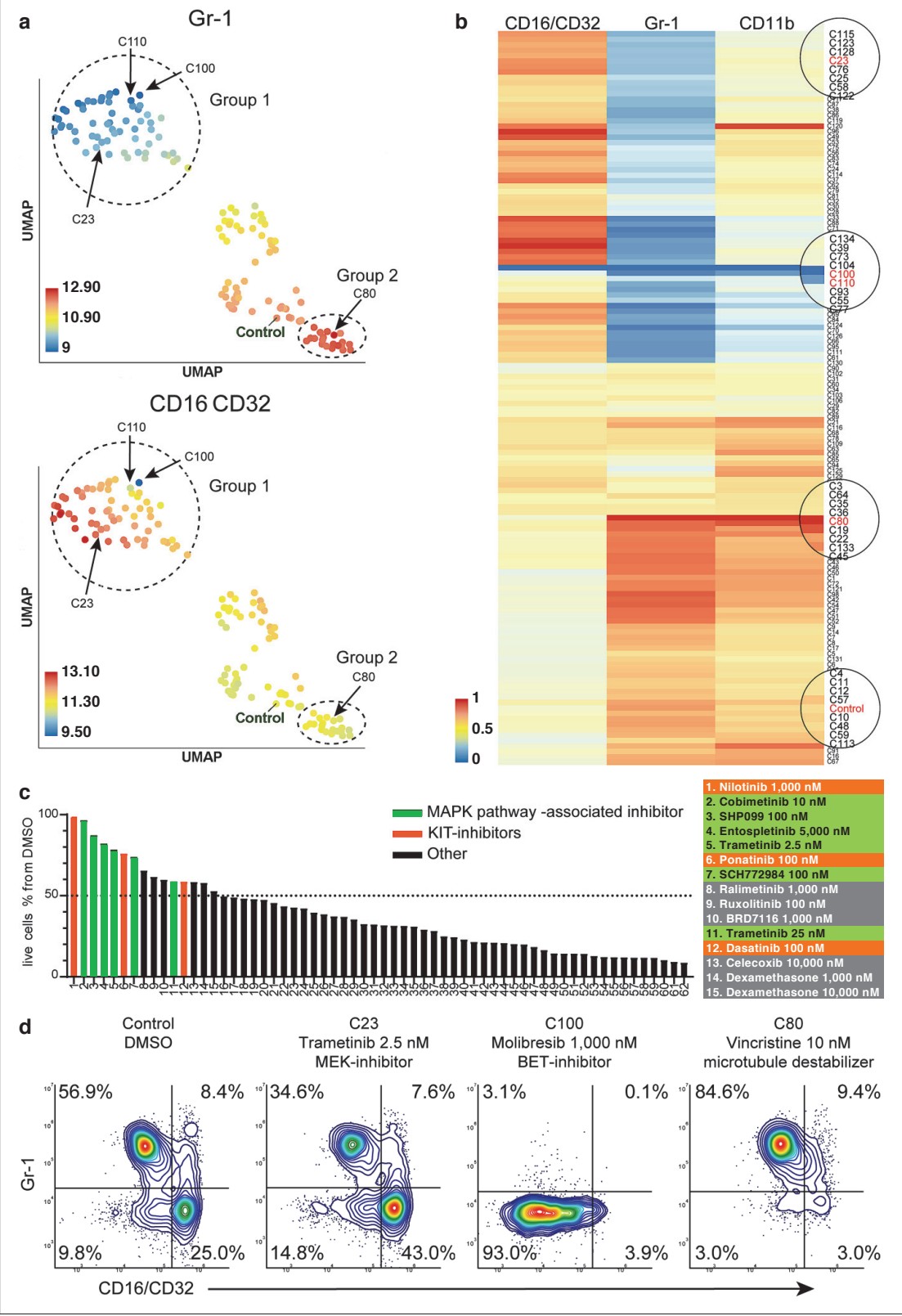

**Figure 5.** compaRe analysis identifies several distinct cell subtype-specific responses in a high-throughput flow cytometry screening of mouse AML cells. (**a**) A UMAP plot of cliques identified by compaRe. Cliques are colored by Gr-1 and CD16/CD32 MFIs. Group one is characterized with reduced Gr-1 and increased CD16/CD32 as compared with control. Group two has increased Gr-1 expression compared with control. (**b**) Heatmap of marker MFIs. Values are normalized between 0 and 1 per marker to make cross-comparisons possible. Cliques containing control, trametinib (2.5 nM) (**C23**), molibresib and

*Figure 5 continued on next page*

*Figure 5 continued*

birabresib (C100 and C110), and vincristine (**C80**) are marked. (**c**) Waterfall plot of compounds belonging to response group 1, showing live cell count as a percentage of control treatment (DMSO). (**d**) Density scatter plots for Control (DMSO), C23, C100, and C80.

Response group 2 consisted of two drugs: birabresib and lenalidomide in different concentrations. These induced a decrease in the MFI of CD45-RA and CD45 channels (*Appendix 1—figure 29a*). In the case of lenalidomide, this response was likely due to cell toxicity and/or growth inhibition (*Appendix 1—figure 29b*). Interestingly, the birabresib response was very pronounced without the

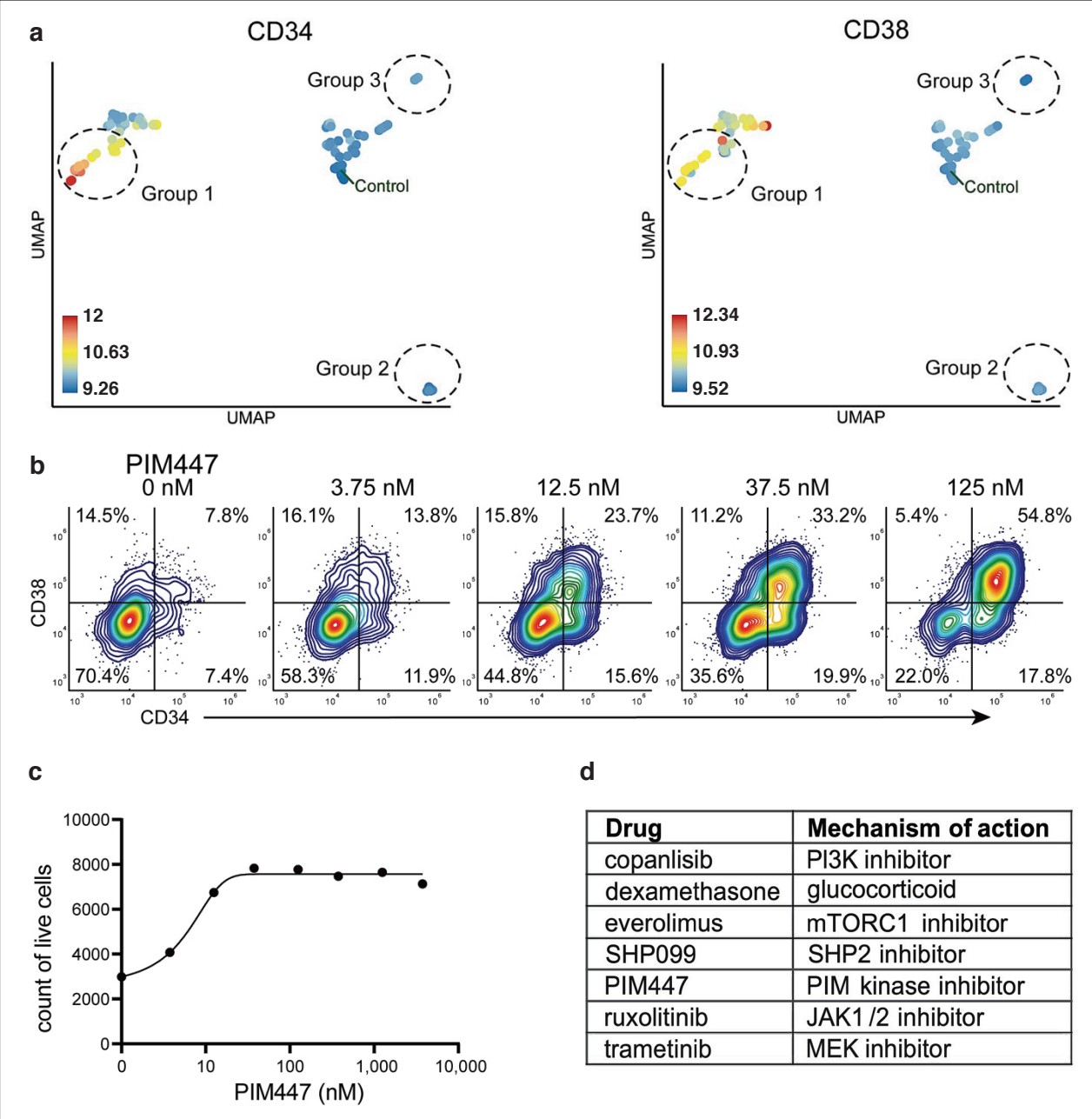

**Figure 6.** Identification of drugs that induce expansion of CD34+/CD38+ cells in an AML patient sample. (**a**) UMAP of cliques identified by compaRe. Cliques are colored by CD34 and CD38 MFIs. Response groups of interest are indicated using a dashed line. (**b**) Example of response group 1: density scatter plot of markers CD34 and CD38 in different concentrations of PIM kinase inhibitor PIM447. (**c**) Count of live cells after 72 hr exposure to different concentrations of PIM kinase inhibitor PIM447. (**d**) Table of drugs that induced expansion of the CD34+/CD38+ cell population.

Chalabi Hajkarim, Karjalainen, *et al*. eLife 2022;11:e73760. DOI: https://doi.org/10.7554/eLife.73760                    11 of 40

loss of live cell numbers, (*Appendix 1—figure 29b*) but with a decrease in the MFI in the cell differentiation marker mix channel (*Appendix 1—figure 29c*).

compaRe also detected response group three as distinct from the controls. This group includes treatment with tretinoin (several concentrations), navitoclax, and mitoxantrone (low dose). Further validation showed the phenotypic response in group three is subtle but with a distinct increase in CD34+ cells (*Appendix 1—figure 29d*). This result highlights compaRe analysis is sensitive enough to identify these subtle changes.

## Discussion

Technological advancements in multiparameter high-throughput screening have enabled testing thousands of biological conditions in a short amount of time. This requires algorithmic development to analyze the large amount of data generated by such technologies. We developed an automated comprehensive toolkit, compaRe, for robust analysis of small- to large-scale multidimensional screening data with several modules for quality control, bias correction, comparison, clustering, and visualization.

The toolkit is unique in many ways. Its quality control and bias correction modules can correct for signal and cell viability drifts in large-scale fluorescence-based screening assays using regression analysis. Its comparison module utilizes a dynamic mass-aware gridding algorithm, which substantially reduces the computing cost and provides robustness to signal shift (batch effect and signal drift). Alternative approaches such as meta-clustering and JSD require both sub-sampling of the data, with the possible loss of valuable subpopulations, and considerably more computing time.

We tested the robustness of the comparison module to batch effect and noise through simulation. The module effectively circumvented the batch effect while JSD and meta-clustering significantly suffered from it. The poor accuracy of meta-clustering demonstrates the drawback of using cluster centroids for similarity comparison across samples while the poor performance of JSD indicates that this approach can work well only in the absence of signal shift. It is of particular note that compaRe does not need subsampling or dimension reduction of the input data.

Multiparameter cytometric analysis of immunophenotypes of AML and MDS patient samples by the comparison module coupled with the EuroFlow AML/MDS antibody panel revealed interpatient heterogeneity and recognizable phenotypic profiles. Even though EuroFlow markers are divided into several discrete tubes, compaRe's comparison module can compare the cell population distribution to measure similarities between patient samples.

We investigated several types of responses evoked by different doses of antineoplastic agents in two high-throughput flow cytometry screening assays of an AML mouse model and an AML human patient. We could identify subtle but distinct phenotypic drug-induced changes. We also identified drugs with different mechanism of action but similar responses. In general, we showed that drugs will cluster together if the cellular processes they affect converge in a specific model.

The quality control and bias correction modules could successfully correct for signal and cell viability drifts in these studies. In our explored assays, signal drift was obviously associated with the order in which wells were read. It was caused by the time differences in antibody incubation across the plate as the high-throughput flow cytometer requires more than one hour to sample all wells in a 384-well plate. For high-density assay plate formats with large numbers of wells, this can cause gradual incremental influences in intensity and cell viability. Therefore, when aligning wells along the order that the flow cytometer sampled the wells, we found a linear trend in MFIs. We benefited from regression analysis to remove the effect of signal shifts.

During the analyses, the compaRe toolkit made it easy to explore and compare highly complex datasets in a substantially reduced timeline. It is equipped with multithreading and can run through command-line interface on a computer server or GUI on a desktop. The GUI provides the investigator with numerous interactive visualization tools including cell staining, graphical representation, and gating. In sum, it provides a total package for fast, accurate, and readily interpretable multiparameter screening data analysis.

## Materials and methods

### Mass cytometry of healthy and pediatric AML bone marrow aspirates

Mass cytometry dataset for 21 samples labeled with 16 surface markers collected from 16 pediatric AML patients obtained at diagnosis and five healthy adult donors (*Levine et al., 2015*) were downloaded from Cytobank Community with the experiment ID 44185. There are 378 FCS files in this experiment with one FCS file for each of 21 patients for each of 17 conditions (two basal replicates and 16 perturbations). All FCS files from a single patient had been pooled then clustered with the PhenoGraph algorithm. Each file includes a column named PhenoGraph that specifies the PhenoGraph cluster to which each event was assigned as an integer. A value of 0 indicates no cluster was assigned because the cells were identified as outliers during some stage of analysis. Using the PhenoGraph column, we determined centroids of cell clusters, and used PhenoGraph to meta-cluster them as described in *Levine et al., 2015* To generate the similarity matrix, we adapted an approach similar to that of compaRe such that each meta-cluster as a spatial unit was treated like a hypercube. We set compaRe's $n$ to four for this assay (Materials and methods and Appendix 1).

### High-throughput flow cytometry of AML mouse model

AML primary splenic cells from Npm1$^{+/cA}$ (*Vassiliou et al., 2011*); Flt3$^{+/ITD}$ (*Lee et al., 2007*); Dnmt3a$^{+/-}$ (*Kaneda et al., 2004*) Mx1-Cre+ (*Kühn et al., 1995*) moribund mice were sorted for c-Kit positivity and expanded ex vivo. AML cells were treated with a library of 116 chemotherapy and immunotherapy antineoplastic agents in a five-point concentration range (*Supplementary file 4*). Treated samples were stained with three informative cell surface antibodies (*Supplementary file 9*) and fluorescence was detected using a high-throughput flow cytometer iQue Screener Plus (Intellicyt). We set compaRe's $n$ to five for this assay.

### High-throughput flow cytometry of an AML human patient sample

Mononuclear cells were isolated from a donated human bone marrow aspirate from an AML patient (Danish National Ethical committee/National Videnskabsetisk Komité permit 1705391). The cells were treated with a library of 40 chemotherapy and targeted antineoplastic agents in a seven-point concentration range (*Supplementary file 7*) for 72 hr. Cells were subsequently incubated with fluorescently labeled antibodies targeting 11 informative cell surface proteins in eight fluorescence channels (*Supplementary file 10*). Samples were read using a high-throughput flow cytometer (iQue Screener Plus, Intellicyt). We set compaRe's $n$ to three for this assay.

### Flow cytometry of AML and MDS patients

Clinical flow cytometry data using a slightly modified AML panel as described by the Euroflow Consortium (*van Dongen et al., 2012*) from 25 bone marrow aspirates from MDS and AML patients from Rigshospitalet (Copenhagen, DK) were used for analysis. Each sample was analyzed using a total of four tubes (Euroflow AML panel tubes 1–4) with eight antibodies in each tube (*Supplementary files 1 and 2*). Acquisition of data was performed on a FACS Canto (Becton Dickinson Immunocytometry Systems), and data analysis was done in the Infinicyt software (Cytognos, Salamanca, Spain). We set compaRe's $n$ to five for this assay.

### Quality control (QC)

Multiwell plate heatmaps of medians come in handy in QC to reveal issues such as signal and cell viability drifts occurring during screening. However, as a typical heatmap has an equally spaced color palette, small but significant differences between wells may be obscured and not visible. Therefore, we normalized the color palette by the distribution of the medians. Also, before clustering, we removed outliers in the negative controls that were different from the others in terms of similarity values measured by compaRe.

### Correcting signal and cell viability drifts

Depending on the protocol by which wells are processed, time may become a major concern so that some specific wells may have lower or higher values than expected. To correct for these sources of bias, we employed a two-step correction: intra-plate shift (signal drift) correction and inter-plate shift

(batch effect) correction. For a given plate, we first fit a linear regression model and then vertically translate points (well values) with respect to the learned line as it rotates to the slope zero. After correcting for the intra-plate bias, the inter-plate bias is corrected by aligning medians of the plates, that is, translating to a common baseline.

## Similarity calculation using dynamic gridding

To measure the similarity between two datasets, compaRe divides each dimension into $n$ subsets for each dataset individually so that a dataset with $d$ dimensions (markers) will be gridded into at most $n^d$ spatial units called hypercubes. compaRe grids only the part of the space encompassing data points, avoiding empty regions. It then measures the proportion of data points for either dataset within each of the corresponding hypercubes. The difference between the two proportions is indicative of the similarity within that relative spatial position represented by each hypercube. The similarity in the exclusive hypercubes is considered 0. We employed local outlier factor (*Breunig et al., 2000*) for anomaly detection and removing noise cells. Averaging these differences across all the hypercubes indicates the amount of similarity between the two datasets.

compaRe captures the configuration of data enabling it to measure similarity even without correcting for signal drift or batch effect (Appendix 1). This way, two technical replicates analyzed by two different instruments or configurations suffering from signal shift will still have the highest similarity. To generate a similarity matrix of multiple input samples, compaRe runs in parallel. The similarity matrix could then be used for identifying clusters of samples such as drugs with similar dose responses.

## Graphical clustering of samples

To cluster samples, we developed a graphical clustering algorithm in which initially all nodes (samples) are connected forming a weighted complete graph wherein edges represent similarity between nodes. This graph is then pruned to remove potential false positive edges for a given cutoff inferred from negative controls. The optimal cutoff turns out to be the minimum weight in the maximum spanning tree of negative control nodes. After pruning, some samples may end up being connected to the negative controls (biologically inactive agents) and some disconnected (active agents). After constructing this graph, clustering is tantamount to finding maximal cliques among potent agents. In addition to maximal cliques, it also reports communities (a clique is a subset of a community). Communities can be seen as loose clusters. In a community, unlike a clique, similarity is not necessarily transitive meaning that if A is similar to B and B is similar to C, A is not necessarily similar to C. If these were three drugs within a community, concluding they had an equal response was not necessarily right unless they would form a clique.

## Dispersion graph and Dispersion map

compaRe visualizes the similarity of samples in the form of a dispersion graph by constructing their maximum spanning tree (Appendix 1, *Appendix 1—figure 30*). compaRe also uses UMAP to represent a dispersion map of clusters. The map is constructed using the centroid (median) of each clique. An informative map shows different groups by coloring the centroids according to their value. These groups are mostly the identified communities the cliques come from.

## Availability of data

Mass cytometry datasets were downloaded from Cytobank Community with the experiment ID 44185. AML mouse and human high-throughput flow cytometry data have been deposited in FLOWRepository with the repository IDs FR-FCM-Z357 and FR-FCM-Z3DP respectively. Flow cytometry data of AML and MDS patients have been deposited in FLOWRepository with the repository ID FR-FCM-Z3ET. Acquisition, installation and more technical details are available in compaRe's online tutorial on (https://github.com/morchalabi/COMPARE-suite, swh:1:rev:df2feaf6aa982e0f6f077eb85f26ac-ce6bb61063, *Chalabi, 2022b*). Similarity measurement and clustering modules as stand-alone tools have been merged into a separate R package and are available for download at (https://github.com/morchalabi/compaRe, swh:1:rev:594106b1e34c17b405064f1a0f9fb39975a4ec79, *Chalabi, 2022a*).

## Acknowledgements

This work was supported through Novo Nordisk Foundation (Novo Nordisk Foundation Center for Stem Cell Biology, DanStem; Grant Number NNF17CC0027852) and Danish Research Center for Precision Medicine in Blood Cancers funded by the Danish Cancer Society (Grant number R223-A13071) and Greater Copenhagen Health Science Partners.

## Additional information

### Funding

| Funder | Grant reference number | Author |
|---|---|---|
| Novo Nordisk Foundation | NNF17CC0027852 | Kirsten Grønbæk<br>Kristian Helin<br>Krister Wennerberg<br>Kyoung-Jae Won |
| Kræftens Bekæmpelse | R223-A13071 | Kirsten Grønbæk<br>Kristian Helin<br>Krister Wennerberg<br>Kyoung-Jae Won |
| Lundbeck Foundation | R313-2019-421 | Kyoung-Jae Won |

The funders had no role in study design, data collection and interpretation, or the decision to submit the work for publication.

### Author contributions

Morteza Chalabi Hajkarim, Conceptualization, Formal analysis, Methodology, Software, Validation, Visualization, Writing – original draft, Writing – review and editing, conceived, designed, and wrote the study with equal contribution; Ella Karjalainen, Conceptualization, Formal analysis, Methodology, Validation, Writing – original draft, Writing – review and editing, conceived, designed, and wrote the study with equal contribution; Mikhail Osipovitch, Software, developed the GUI; Konstantinos Dimopoulos, Data curation, Validation, assembled and annotated clinical cytometry data and assisted in its analysis; Sandra L Gordon, Data curation, Formal analysis, Validation, Writing – original draft, designed and completed the AML patient sample drug screening, Writing – review and editing; Francesca Ambri, Data curation, Formal analysis, Validation, designed and completed the AML patient sample drug screening; Kasper Dindler Rasmussen, Methodology, generated the mouse cell models, Writing – review and editing; Kirsten Grønbæk, Funding acquisition, Supervision, assembled and annotated clinical cytometry data and assisted in its analysis, Writing – review and editing; Kristian Helin, Funding acquisition, Supervision, generated the mouse cell models; Krister Wennerberg, Kyoung-Jae Won, Conceptualization, Funding acquisition, Investigation, Methodology, Project administration, Supervision, Validation, Writing – original draft, conceived, designed, and wrote the study with equal contribution, Writing – review and editing

### Author ORCIDs

Morteza Chalabi Hajkarim ⓘ http://orcid.org/0000-0002-2039-2676
Ella Karjalainen ⓘ http://orcid.org/0000-0002-9865-5384
Sandra L Gordon ⓘ http://orcid.org/0000-0003-0270-8291
Francesca Ambri ⓘ http://orcid.org/0000-0002-1999-9294
Kasper Dindler Rasmussen ⓘ http://orcid.org/0000-0002-7344-4177
Kirsten Grønbæk ⓘ http://orcid.org/0000-0002-1535-9601
Krister Wennerberg ⓘ http://orcid.org/0000-0002-1352-4220
Kyoung-Jae Won ⓘ http://orcid.org/0000-0002-2924-9630

### Ethics

Human subjects: The informed consent, and consent to publish of patient samples in this study has been approved by the Danish National Science Ethics Committee/National Videnskabsetisk Komite: Målrettet behandling af patienter med blodsygdomme, license no. 1705391.

### Decision letter and Author response

Decision letter https://doi.org/10.7554/eLife.73760.sa1

Author response https://doi.org/10.7554/eLife.73760.sa2

## Additional files

### Supplementary files

- Supplementary file 1. EuroFlow antibody panel for AML and MDS.
- Supplementary file 2. Antibodies used by compaRe in the AML/MDS study.
- Supplementary file 3. Clinical reports of the patients in the AML/MDS study.
- Supplementary file 4. Drug panel of the AML mouse model study.
- Supplementary file 5. Drug clusters identified by compaRe in the AML mouse model study.
- Supplementary file 6. Mechanism of action of the response group one in *Figure 5c*.
- Supplementary file 7. Drug panel of the AML human sample study.
- Supplementary file 8. Drug clusters identified by compaRe in the AML human sample study.
- Supplementary file 9. Antibodies used by compaRe in the AML mouse model study.
- Supplementary file 10. Antibodies used by compaRe in the AML human model study.
- Transparent reporting form

### Data availability

Mass cytometry datasets were downloaded from Cytobank Community with the experiment ID 44185. AML mouse and human high-throughput flow cytometry data have been deposited in FLOWRepository with the repository IDs FR-FCM-Z357 and FR-FCM-Z3DP respectively. Flow cytometry data of AML and MDS patients have been deposited in FLOWRepository with the repository ID FR-FCM-Z3ET. Acquisition, installation and more technical details are available in compaRe's online tutorial on (https://github.com/morchalabi/COMPARE-suite, (copy archived at swh:1:rev:df2feaf6aa982e0f-6f077eb85f26acce6bb61063)). Similarity measurement and clustering modules as stand-alone tools have been merged into a separate R package and are available for download at (https://github.com/morchalabi/compaRe, (copy archived at swh:1:rev:594106b1e34c17b405064f1a0f9fb39975a4ec79)).

The following datasets were generated:

| Author(s) | Year | Dataset title | Dataset URL | Database and Identifier |
|---|---|---|---|---|
| Morteza C H, Ella K, Mikhail O, Konstantinos D, Sandra L G, Francesca A, Kasper D R, Kirsten G, Kristian H, Krister W, Kyoung-Jae W | 2021 | AML/MDS Flow Cytometry | https://flowrepository.org/id/FR-FCM-Z3ET | Flowrepository, FR-FCM-Z3ET |
| Morteza C H, Ella K, Mikhail O, Konstantinos D, Sandra L G, Francesca A, Kasper D R, Kirsten G, Kristian H, Krister W, Kyoung-Jae W | 2021 | AML Mouse High-throughput Flow Cytometry | https://flowrepository.org/id/FR-FCM-Z357 | Flowrepository, FR-FCM-Z357 |
| Morteza C H, Ella K, Mikhail O, Konstantinos D, Sandra L G, Francesca A, Kasper D R, Kirsten G, Kristian H, Krister W, Kyoung-Jae W | 2021 | AML Human High-throughput Flow Cytometry | https://flowrepository.org/id/FR-FCM-Z3DP | Flowrepository, FR-FCM-Z3DP |
| Morteza CH | 2022 | Comprehensive and unbiased multiparameter high-throughput screening by compaRe finds effective and subtle drug responses in AML models | https://github.com/morchalabi/compaRe | GitHub, github.com/morchalabi/compaRe |

The following previously published dataset was used:

| Author(s) | Year | Dataset title | Dataset URL | Database and Identifier |
|---|---|---|---|---|
| Levine JH, Simonds EF, Bendall SC, Davis KL, Amir el AD, Tadmor MD | 2015 | Data-Driven Phenotypic Dissection of AML Reveals Progenitor-like Cells that Correlate with Prognosis | https://premium.cytobank.org/cytobank/experiments | Cytobank, 44185 |

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

## Appendix 1

### High-throughput flow cytometry of AML mouse model

Leukemic spleen cells were sorted for c-Kit positivity from Npm1$^{+/cA}$; Flt3$^{+/ITD}$; Dnmt3a$^{+/-}$; Mx1-Cre+ moribund mice. Shortly, c-Kit+ splenic cells were expanded for two passages in StemPro-34 SFM media (Gibco) with 100 μM 2-Mercaptoethanol (Gibco), 20 ng/ml murine SCF, 10 ng/ml murine IL-3 and 10 ng/ml IL-6 added (Peprotech), with complete media change every two/three days. Aliquots of one million cells were frozen down in 90% media 10% DMSO. Frozen aliquots were taken up and expanded for one week before drug screening. 5000 cells in 25 μl of media per well was seeded into 384-well plates (Greiner) containing a library of 116 compounds (*Supplementary file 4*) in a five-point concentration range. After 72 h incubation at 37 °C, 15 μl of medium was aspirated from each well and antibodies (*Supplementary file 9*) were added to drug plates using acoustic dispensing. Plates were incubated 40 min at RT, covered from light. Next, dead cell dye 7-AAD (BD) was added, and samples were read using a high-throughput flow cytometer iQue Screener Plus (Intellicyt). To remove noise from the data by excluding the most broadly toxic treatments, doublets and dead cells were omitted (*Appendix 1—figure 31*) and only samples with at least 1,000 live cells were selected for further analyses (selected 465 wells out of 600).

### High-throughput flow cytometry of human AML

Donated MNCs from human bone marrow aspirates (Danish National Ethical committee/National Videnskabsetisk Komité permit 1705391) were thawed and allowed to rest overnight in assay media: StemSpan II-SFEM (StemCell), 100 U/ml penicillin/streptomycin (Thermo), including the following human recombinant cytokines from Preprotech (unless otherwise stated), 50 ng/ml Flt3 ligand (StemCell), 10 ng/ml IL3, 10 ng/ml IL-1beta, 20 ng/ml IL6, 20 ng/ml G-CSF, 20 ng/ml GM-CSF, and 10 ng/ml SCF, and the following compounds diluted in DMSO (Merck) 1 μM UM729 (Selleckchem) and 500 nM StemRegenin-1 (MedChemExpress). Before being counted and re-suspended in fresh assay media at a density of $5 \times 10^5$ cells/ml. A 20 μl/well was plated in 384-well conical bottom plates (Greiner Bio-One) containing 25 nl of compounds (*Supplementary file 7*) in DMSO. After 72 hr incubation at 37 °C, 95% RH, 5% $CO_2$ antibodies and viability dye were added to the plates using acoustic dispensing (Echo, Labcyte). Plates were incubated for 1.5 hr covered from light at RT. The samples were then run on an iQue Screener Plus (Intellicyt) high-throughput flow cytometer. The data was gated to remove noise, doublets, and dead cells (*Appendix 1—figure 30*). The antibodies and stains used are described in *Supplementary file 10*.

### Signal and cell viability drifts correction in compaRe

To correct signal drift, we employed a two-step correction: intra-plate correction and inter-plate correction. For a given plate, we first fit a linear regression model and then vertically translate points (MFIs) with respect to the leaned line as it rotates to slope zero. This is because the relative distance between the points must be retained as much as possible, and no point must be translated to $x^+y^-$ quadrant after correction. To make sure the learned line is not affected by outliers, we first removed them using the interquartile range. In this way, a point at $(y, x)$ is translated to $\left( y\frac{b}{mx+b}, x \right)$ after intra-plate correction. The correction coefficient $\frac{b}{mx+b}$ derives from the ratio of y-coordinates of any point on the regression line before and after translation: $\frac{y^*}{y} = \frac{b}{mx+b}$ where $y^*$ is translated $y$, $m$ is the slope and $b$ is the intercept of the line. This ratio holds true for all other points in the $xy$-plane.

After correcting for intra-plate signal drift, inter-plate signal drift is corrected by aligning MFI medians of the plates, that is, translating to a common baseline. Let $b^*$ be the baseline, and $b$ be the median of corrected MFIs in a plate, then the inter-plate correction coefficient is given by $\frac{b^*}{b}$ , and a point at $(y, x)$ is translated to $\left( y\frac{b^*}{b}, x \right)$ . The same approach is employed for correcting cell viability bias (*Appendix 1—figure 28*).

### Similarity measurement in compaRe

compaRe can measure the similarity between two datasets with many variables (dimensions) and observations (data points). compaRe divides each dimension into $n$ subsets so that a dataset with $d$ dimensions will be divided into at most $n^d$ spatial units called hypercubes. The hypercubes are formed for either dataset individually. It, then measures the proportion of the observations within each of the corresponding hypercubes. The difference between the two proportions is indicative of the similarity within that relative spatial position represented by that hypercube so that for two similar

datasets this difference is near zero in the majority of the hypercubes. Averaging these differences across all the hypercubes indicates the amount of similarity between the two datasets.

It is important to compare two samples across their corresponding hypercubes representing the same relative spatial positions. This means a universal numbering rule is required to ensure having corresponding hypercubes for the two samples in the end. This problem can be modeled as a tree that at each level $l$ (dimension) grows $n^l$ new branches (divisions) (**Appendix 1—figure 32**). However, as the number of branches increases exponentially with $l$, implementing the tree is infeasible. To overcome this problem, we instead employed a dynamic algorithm in which the hypercube number of each observation is dynamically updated at each iteration. In this approach, the child node number must be found from its parent's, that is, previous iteration.

Rewriting the branch numbers to include more information reveals that if $r_{l-1} = \left( n^0 + \ldots + n^{l-2} \right) + f_{l-1} + s_{l-1} n^{l-2}$ is the parent node's number, the child node's number will be $r_l = \left( n^0 + \ldots + n^{l-2} + n^{l-1} \right) + \left( n f_{l-1} + s_{l-1} \right) + s_l n^{l-1}$ where $l$ is the child's level, $f_l = 0, \ldots, n^{l-1} - 1$ is the number of families behind, and $s_l = 0, \ldots, l - 1$ is the number of siblings behind. Therefore, to find child node $r_l$, we first need to calculate $f_{l-1}$ and $s_{l-1}$ of its parent as follows:

$$ s_{l-1} = \left\lfloor \frac{ r_{l-1} - \left( n + \ldots + n^{l-2} \right) }{ n^{l-2} } \right\rfloor \tag{1} $$

$$ f_{l-1} = r_{l-1} - \left( n^0 + \ldots + n^{l-2} \right) - s_{l-1} n^{l-2} \tag{2} $$

It can be noticed that $r_{l-1}$ and $s_l$ are always known, $f_l = n f_{l-1} + s_{l-1}$, and $\left( n^0 + \ldots + n^{l-1} \right) - 1$ is actually the largest node number at the $l$th level. Therefore, the problem we need to dynamically solve for each child at each dimension as the tree grows is:

$$ r_l = \left( n^0 + \ldots + n^{l-1} \right) + f_l + s_l n^{l-1} \tag{3} $$

Since the similarity metric decreases for each exclusive hypercube, it is important to rid the two samples of outliers lying significantly far from the subpopulations of observations. However, at the same time we need to make sure smaller subpopulations (like rare cell subpopulations) are not mistaken for outliers. We employed local outlier factor which is a powerful tool for anomaly detection. **Figure 1** shows an actual AML dataset with three surface markers dissected by compaRe wherein each distinct color corresponds to data points within one abstract hypercube.

compaRe captures the morphology of high dimensional data enabling it to measure similarity even in the presence of moderate signal shift. For example, two technical replicates analyzed by two different instruments or configurations suffering from signal shift will still have the highest similarity by compaRe unless the shift is severe or has modified the morphology of the cell populations which practically does not happen as a result of batch effect or signal drift. This strategy helps compaRe circumvent signal drift or batch effect left uncorrected. Considering that any signal drift correction is essentially an approximate method, this feature is an advantage for compaRe, because together with the correction method they create a synergistic effect.

compaRe is a mass-aware approach meaning it forms hypercubes only around concentrations of data points avoiding areas which are devoid of data points. This substantially speeds up the process by saving a lot of CPU time and memory space making it feasible to compare datasets with numerous variables. As an example, dividing each dimension blindly into just three regions yields more than 1.5 billion regions for consideration for a dataset with as few as 19 surface markers. In practice, however, it turns out many of these regions are empty so using a mass-aware gridding instead of blind gridding improves the comparison complexity from $\Theta \left( n^d \right)$ to $O \left( n^d \right)$ (asymptotic notations to represent algorithmic complexity). Even if no region is empty, since compaRe benefits from dynamic programming, it can still finish the process quite fast. Changing $n$ tunes the level of smoothing so that a value between 3–5 works for most assays.

Dynamic programming is key for reducing processing power. In general, the goal is to bin/grid data into relative expression groups (hypercubes). Gridding can be implemented by a simple algorithm dividing each dimension in each iteration. However, as pointed out above, after a couple of rounds, this naïve algorithm turns out to be infeasible. Therefore, one need a more efficient

algorithm for implementing gridding. Dynamic programming turned out to be quite effective. What makes dynamic programming very effective is its ability to memorize the values computed in the previous iterations avoiding recomputing potentially expensive algebraic operations (Appendix 1-equation 3).

To generate a similarity matrix of multiple input samples, compaRe runs in parallel for the samples in the upper-triangular submatrix using a multithreading approach. The similarity matrix could then be used for identifying clusters of samples such as drugs with similar dose responses like predicting the mechanism of action of drugs in development.

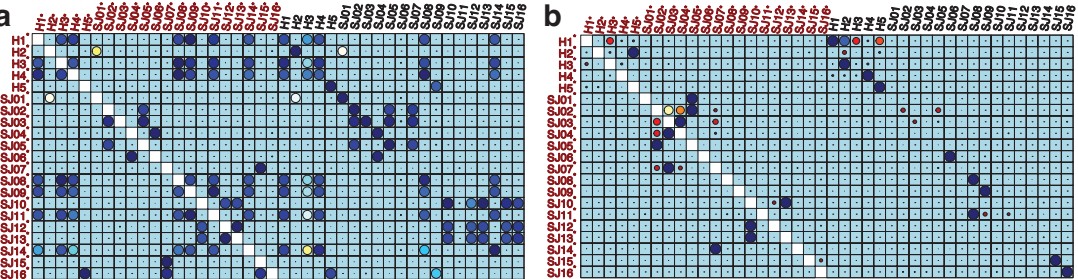

**Appendix 1—figure 1.** Performance of meta-clustering with SPADE FlowSOM in the presence of batch effect. Similarity matrices generated by FlowSOM and SPADE are shown in (**a**) and (**b**) respectively. Size and color of dots represent the level of similarity. Self-comparisons were removed. Noise was added (marked with *) to the original 21 mass cytometry samples of bone marrow aspirates from 16 pediatric AML patients (**S**) and five healthy adult donors (**H**).

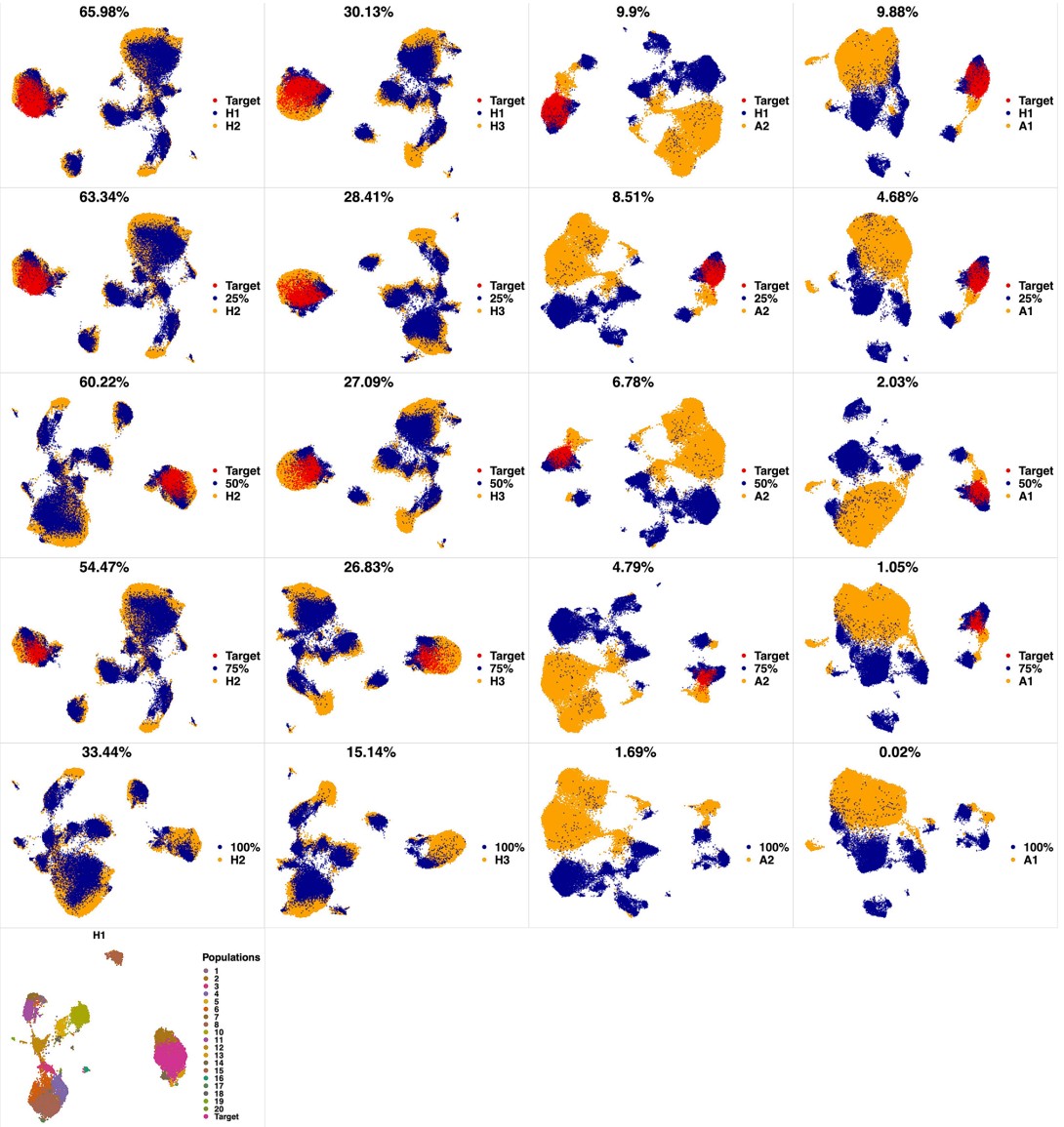

**Appendix 1—figure 2.** Phenotypic characterization in a high-parameter heterogeneous population of cell types. Cells from a target cluster (an immunophenotypic cell population) were gradually removed to contort its configuration. We used a dataset of 3 healthy and two pediatric AML bone marrow mononuclear cell samples from the data provided in the 6th reference. Samples were stained with 29 (15 membrane and 14 intracellular signaling) markers. Taking H1 as reference, we gradually removed 25%, 50%, 75% and 100% (phenotypic changes) of cells from the target cluster identified by PhenoGraph. To capture the higher heterogeneity harbored in the AML samples, we set compaRe's *n* to 4 while we set it to 3 for healthy samples. Each column was scaled individually retaining mutual differences.

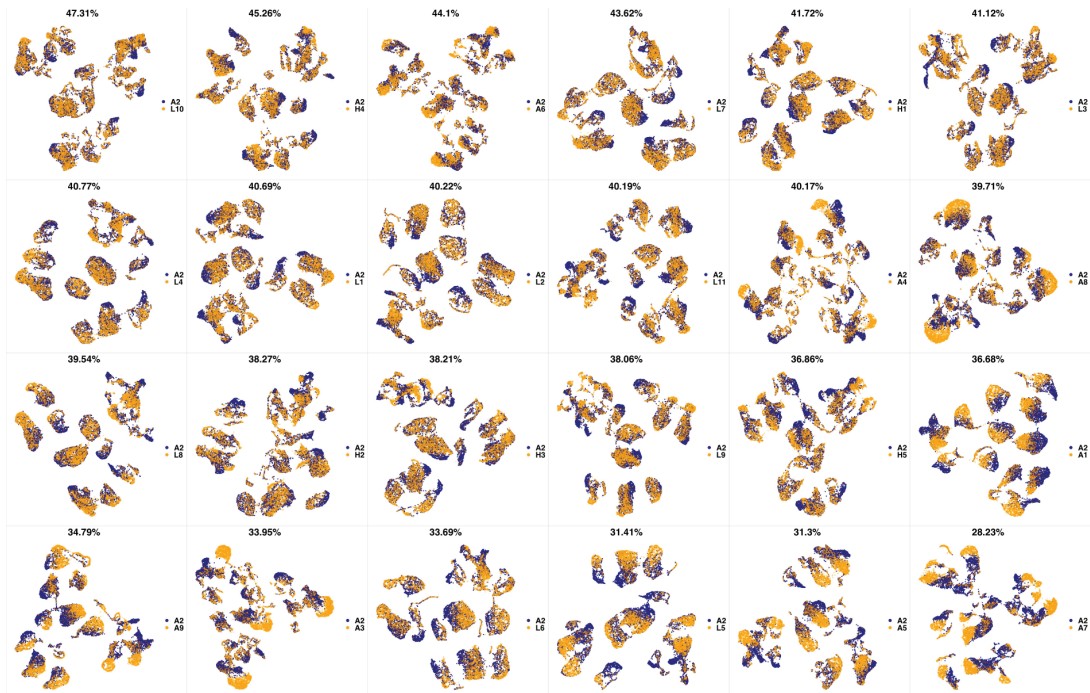

**Appendix 1—figure 3.** UMAP projections of A2 sample against all other patient samples. From top left to bottom right, the similarity measured by compaRe decreases as the degree of overlap decreases and the number of exclusive cell populations increases.

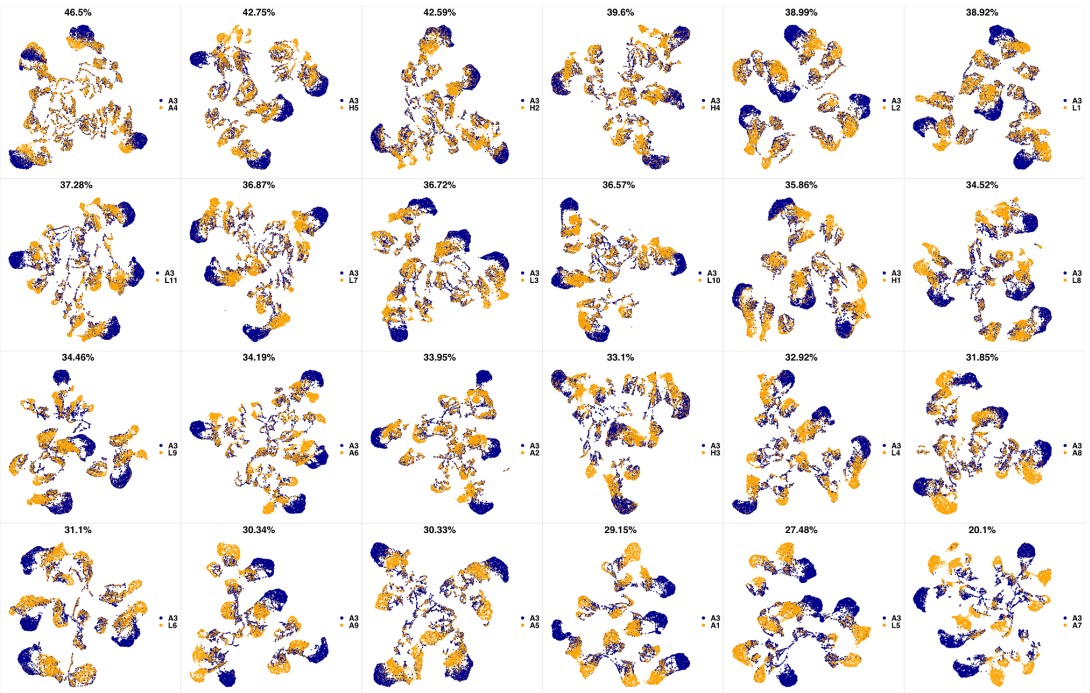

**Appendix 1—figure 4.** UMAP projections of A3 sample against all other patient samples. From top left to bottom right, the similarity measured by compaRe decreases as the degree of overlap decreases and the number of exclusive cell populations increases.

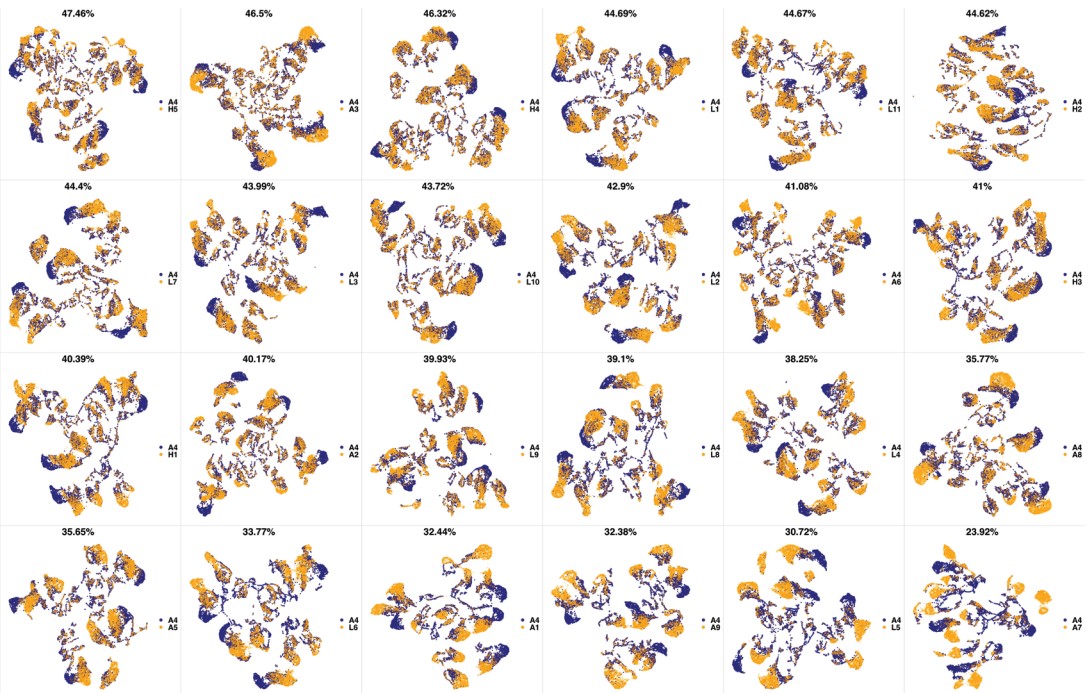

**Appendix 1—figure 5.** UMAP projections of A4 sample against all other patient samples. From top left to bottom right, the similarity measured by compaRe decreases as the degree of overlap decreases and the number of exclusive cell populations increases.

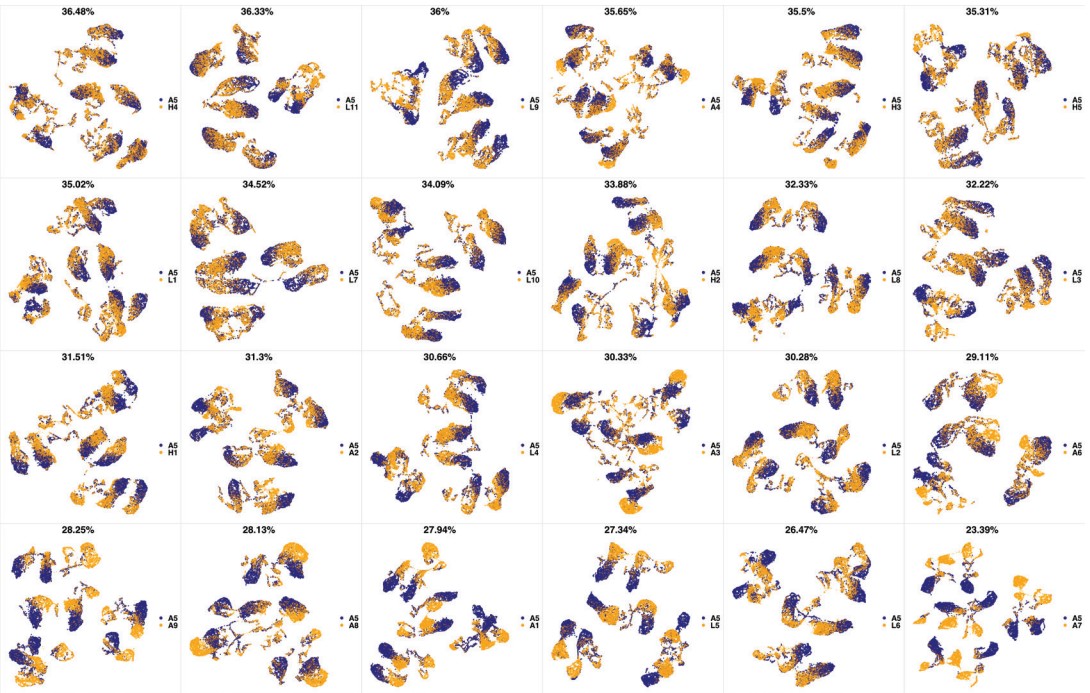

**Appendix 1—figure 6.** UMAP projections of A5 sample against all other patient samples. From top left to bottom right, the similarity measured by compaRe decreases as the degree of overlap decreases and the number of exclusive cell populations increases.

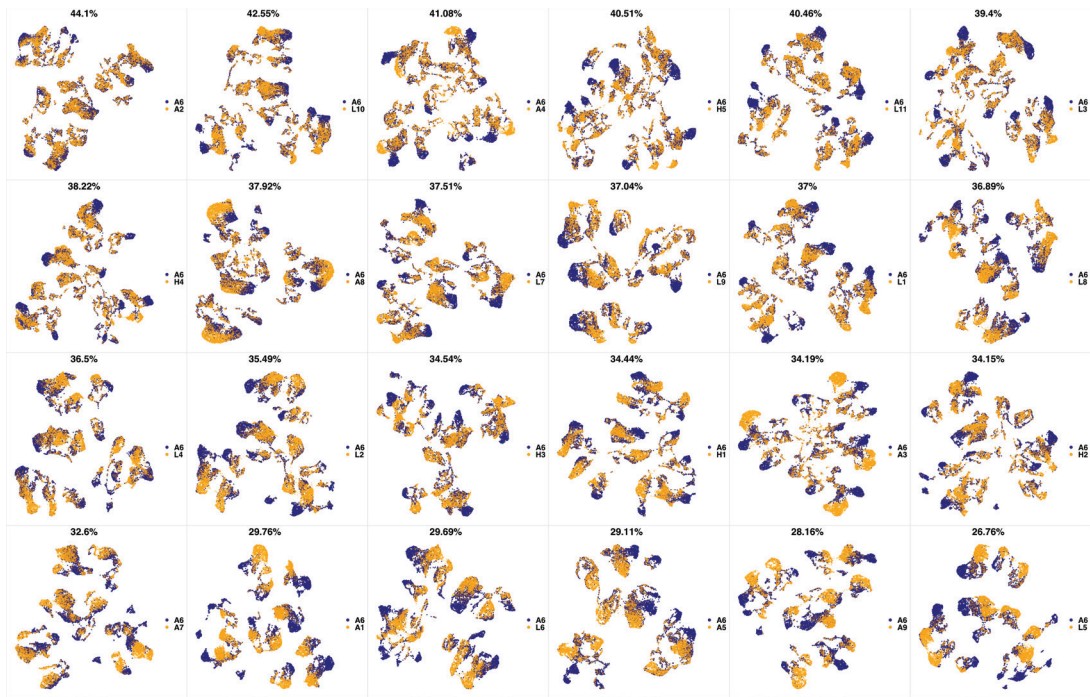

**Appendix 1—figure 7.** UMAP projections of A6 sample against all other patient samples. From top left to bottom right, the similarity measured by compaRe decreases as the degree of overlap decreases and the number of exclusive cell populations increases.

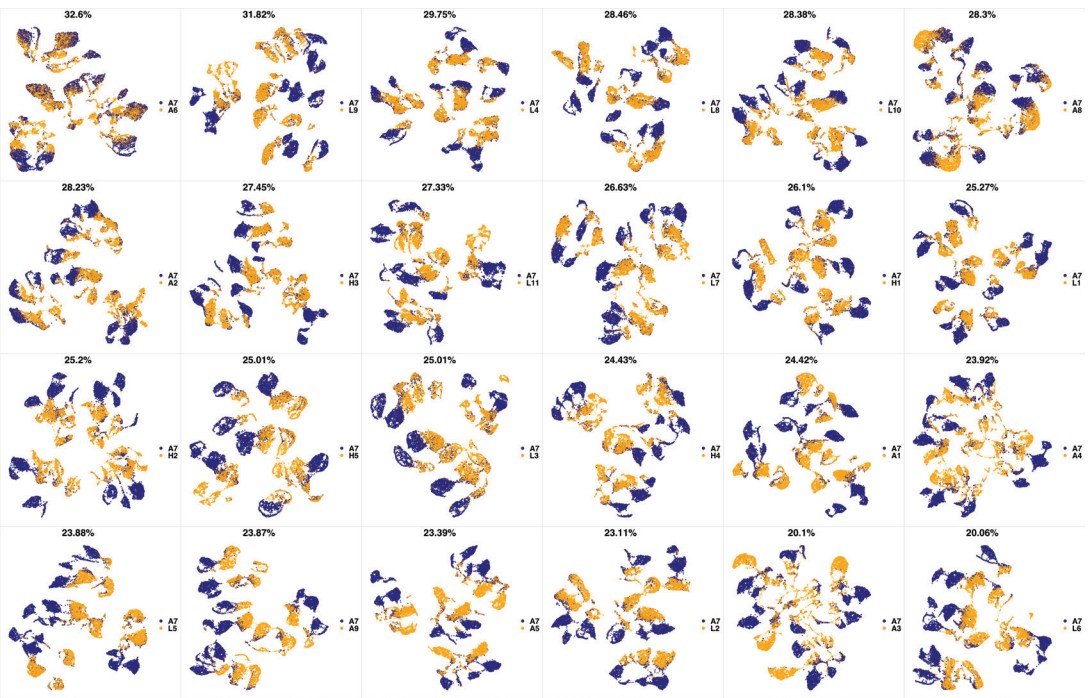

**Appendix 1—figure 8.** UMAP projections of A7 sample against all other patient samples. From top left to bottom right, the similarity measured by compaRe decreases as the degree of overlap decreases and the number of exclusive cell populations increases.

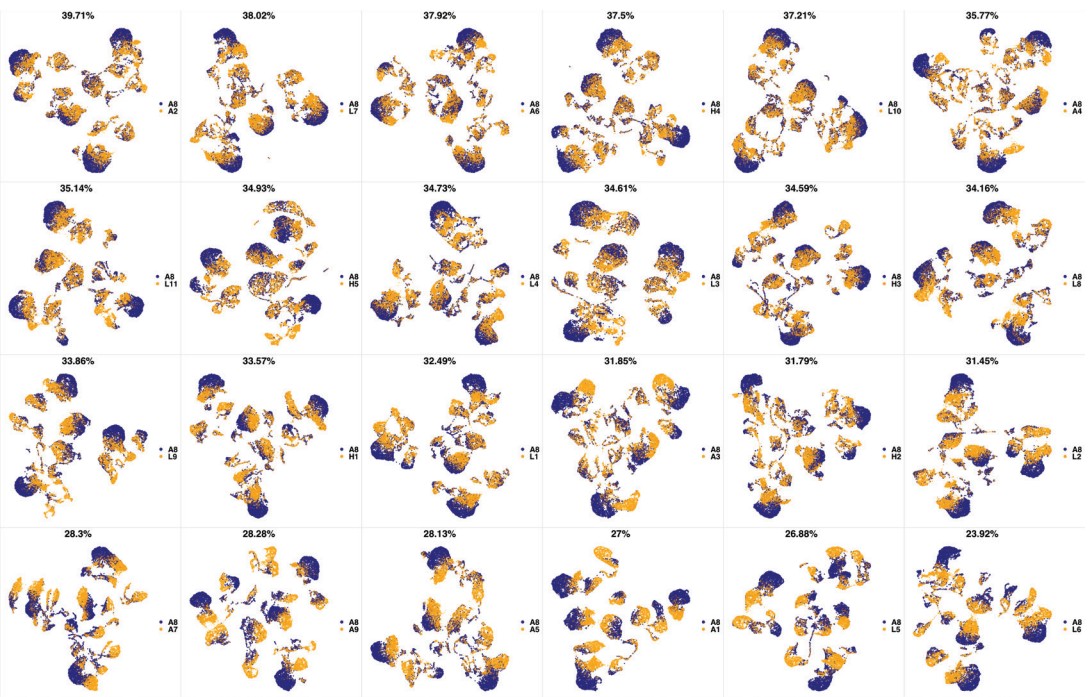

**Appendix 1—figure 9.** UMAP projections of A8 sample against all other patient samples. From top left to bottom right, the similarity measured by compaRe decreases as the degree of overlap decreases and the number of exclusive cell populations increases.

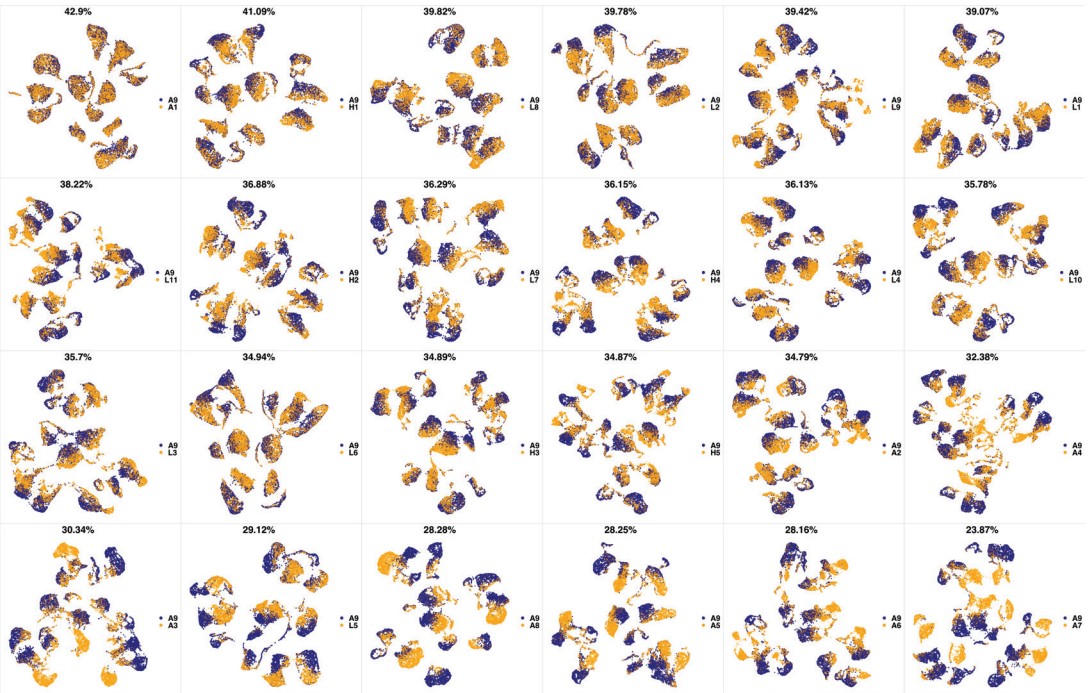

**Appendix 1—figure 10.** UMAP projections of A9 sample against all other patient samples. From top left to bottom right, the similarity measured by compaRe decreases as the degree of overlap decreases and the number of exclusive cell populations increases.

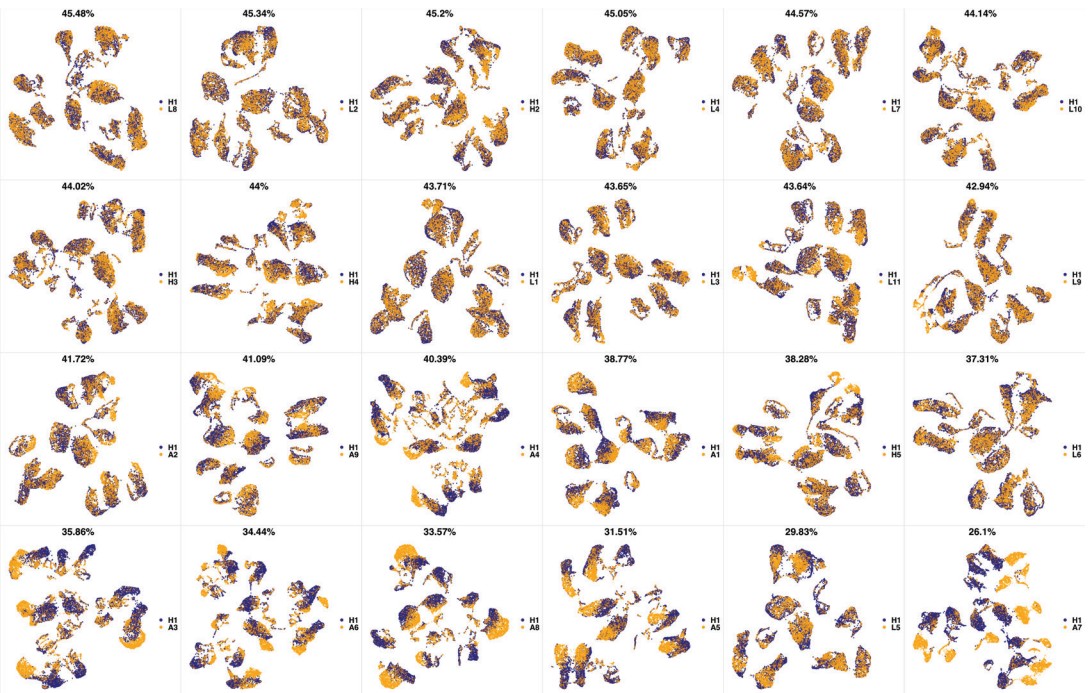

**Appendix 1—figure 11.** UMAP projections of H1 sample against all other patient samples. From top left to bottom right, the similarity measured by compaRe decreases as the degree of overlap decreases and the number of exclusive cell populations increases.

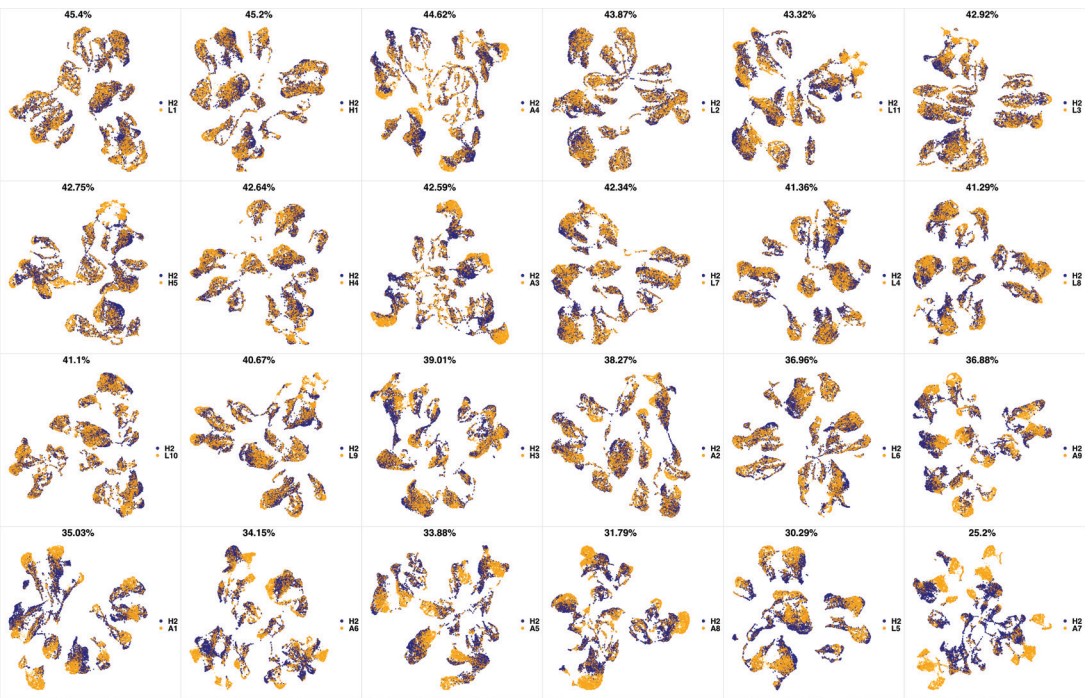

**Appendix 1—figure 12.** UMAP projections of H2 sample against all other patient samples. From top left to bottom right, the similarity measured by compaRe decreases as the degree of overlap decreases and the number of exclusive cell populations increases.

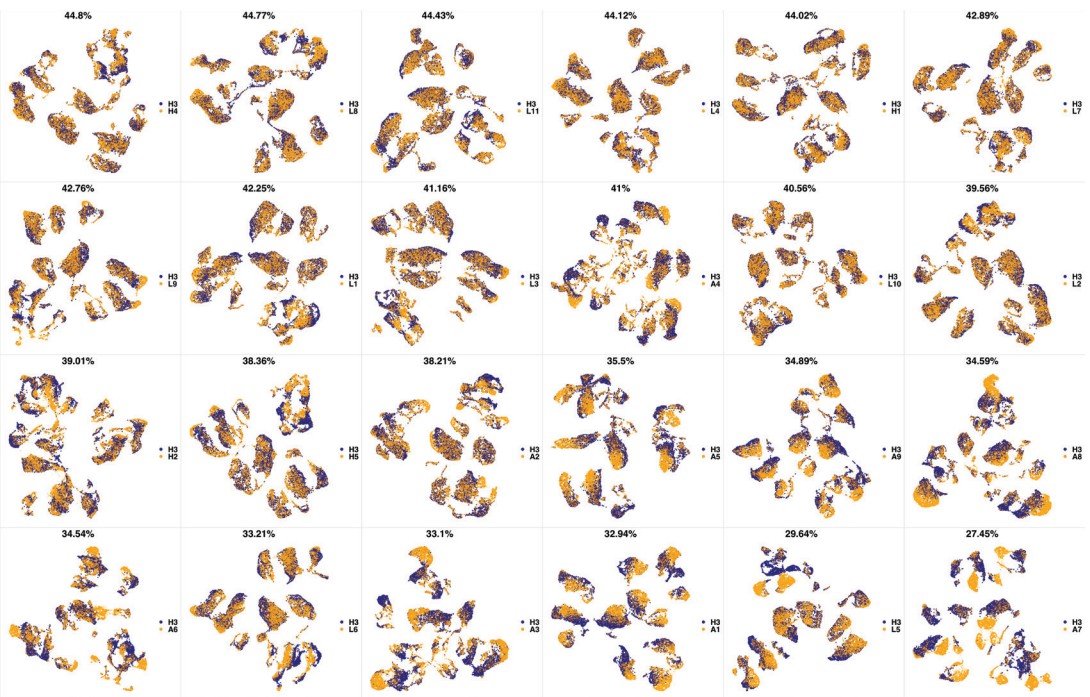

**Appendix 1—figure 13.** UMAP projections of H3 sample against all other patient samples. From top left to bottom right, the similarity measured by compaRe decreases as the degree of overlap decreases and the number of exclusive cell populations increases.

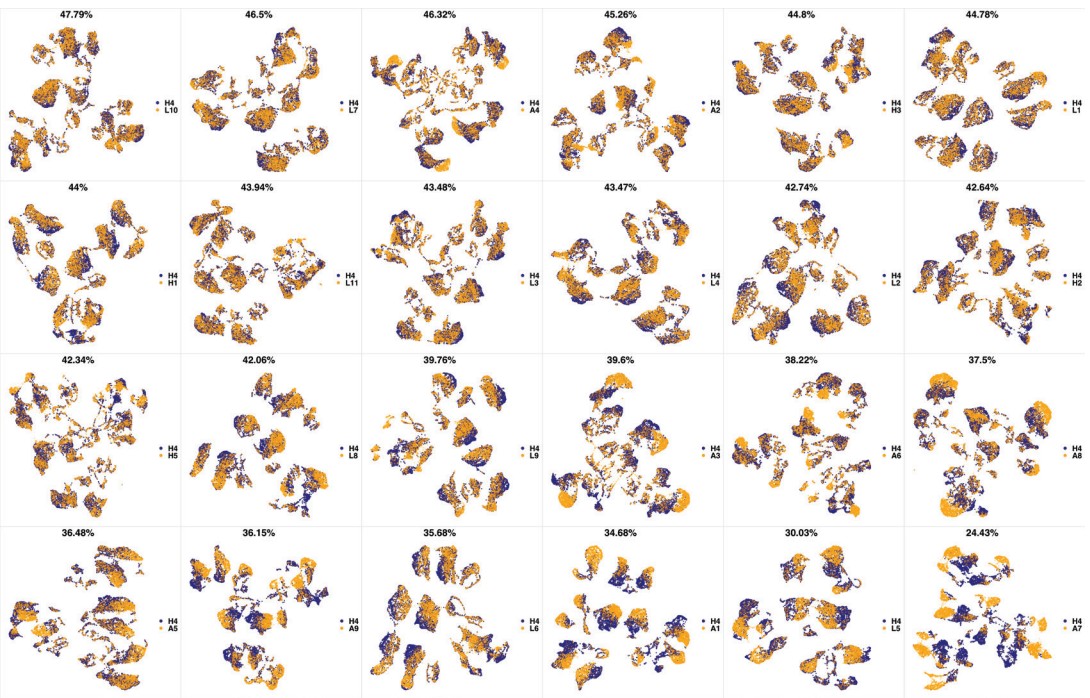

**Appendix 1—figure 14.** UMAP projections of H4 sample against all other patient samples. From top left to bottom right, the similarity measured by compaRe decreases as the degree of overlap decreases and the number of exclusive cell populations increases.

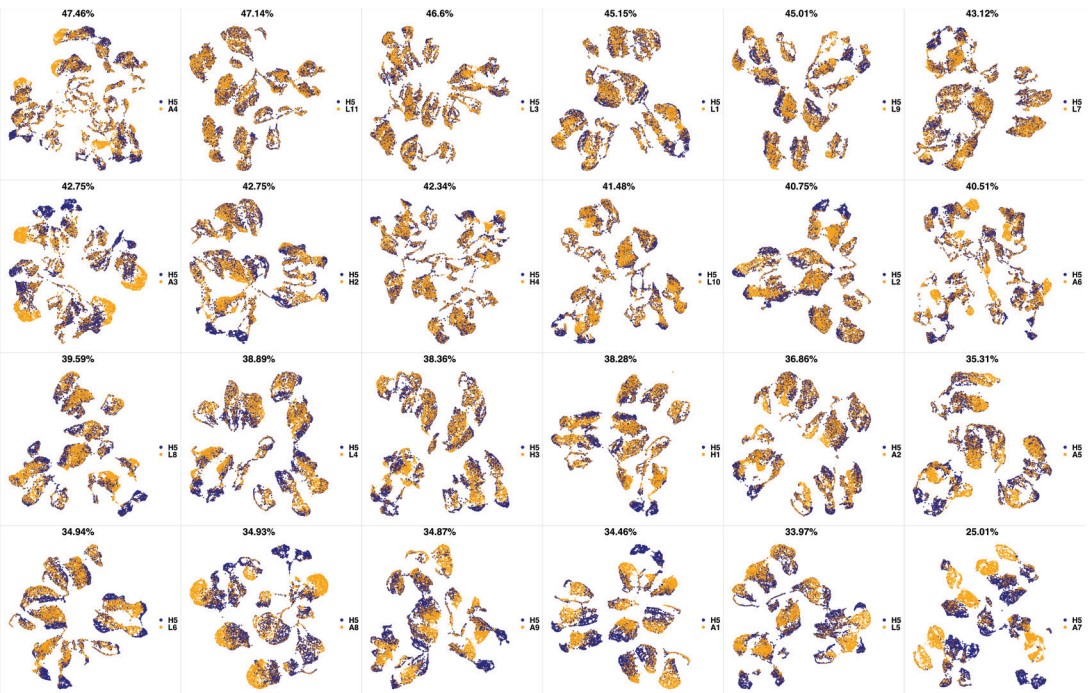

**Appendix 1—figure 15.** UMAP projections of H5 sample against all other patient samples. From top left to bottom right, the similarity measured by compaRe decreases as the degree of overlap decreases and the number of exclusive cell populations increases.

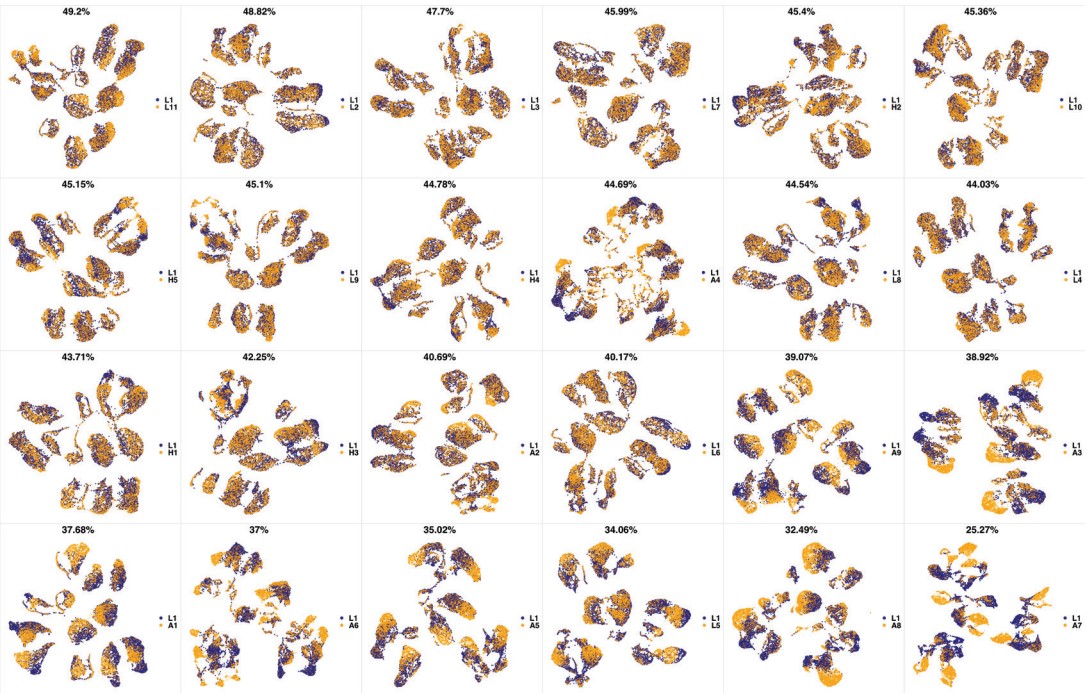

**Appendix 1—figure 16.** UMAP projections of L1 sample against all other patient samples. From top left to bottom right, the similarity measured by compaRe decreases as the degree of overlap decreases and the number of exclusive cell populations increases.

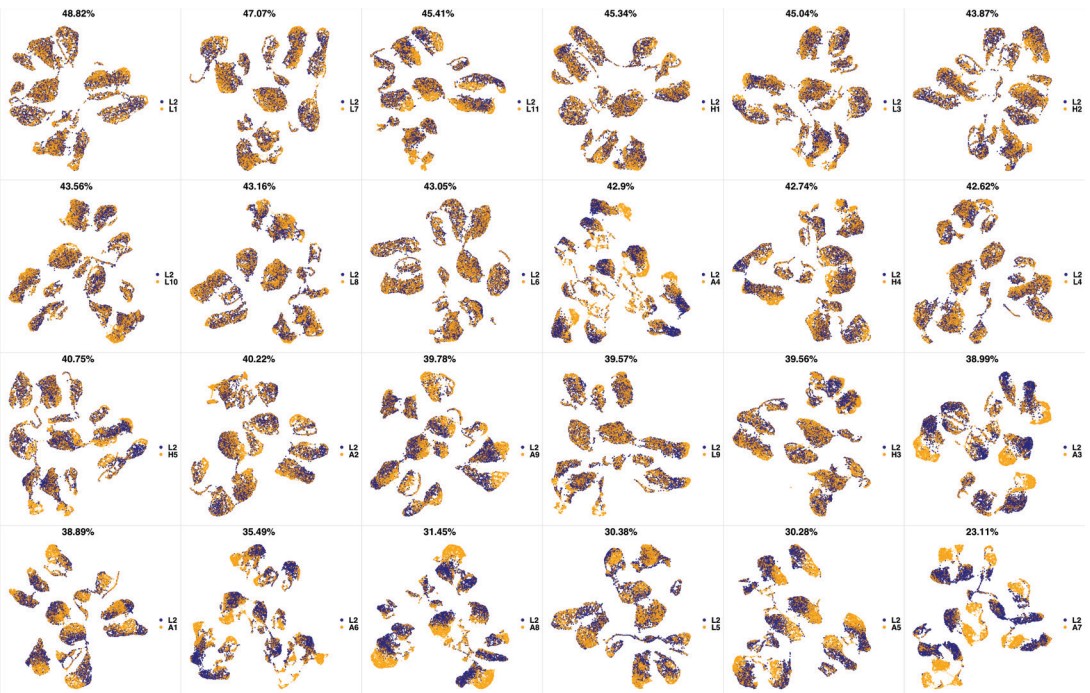

**Appendix 1—figure 17.** UMAP projections of L2 sample against all other patient samples. From top left to bottom right, the similarity measured by compaRe decreases as the degree of overlap decreases and the number of exclusive cell populations increases.

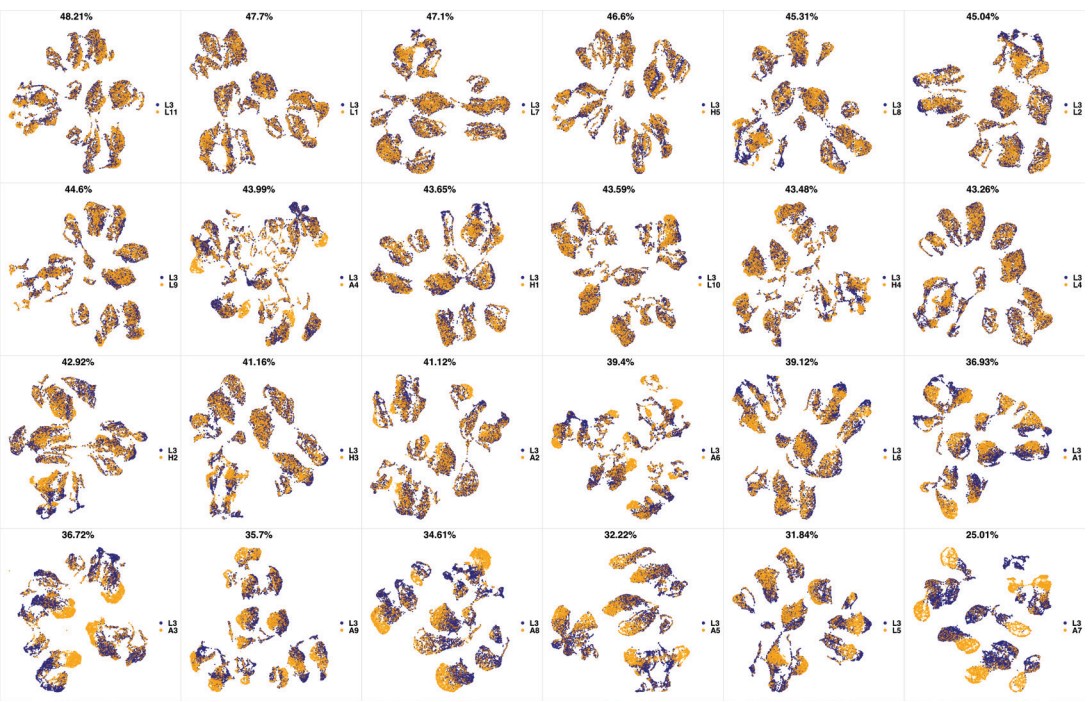

**Appendix 1—figure 18.** UMAP projections of L3 sample against all other patient samples. From top left to bottom right, the similarity measured by compaRe decreases as the degree of overlap decreases and the number of exclusive cell populations increases.

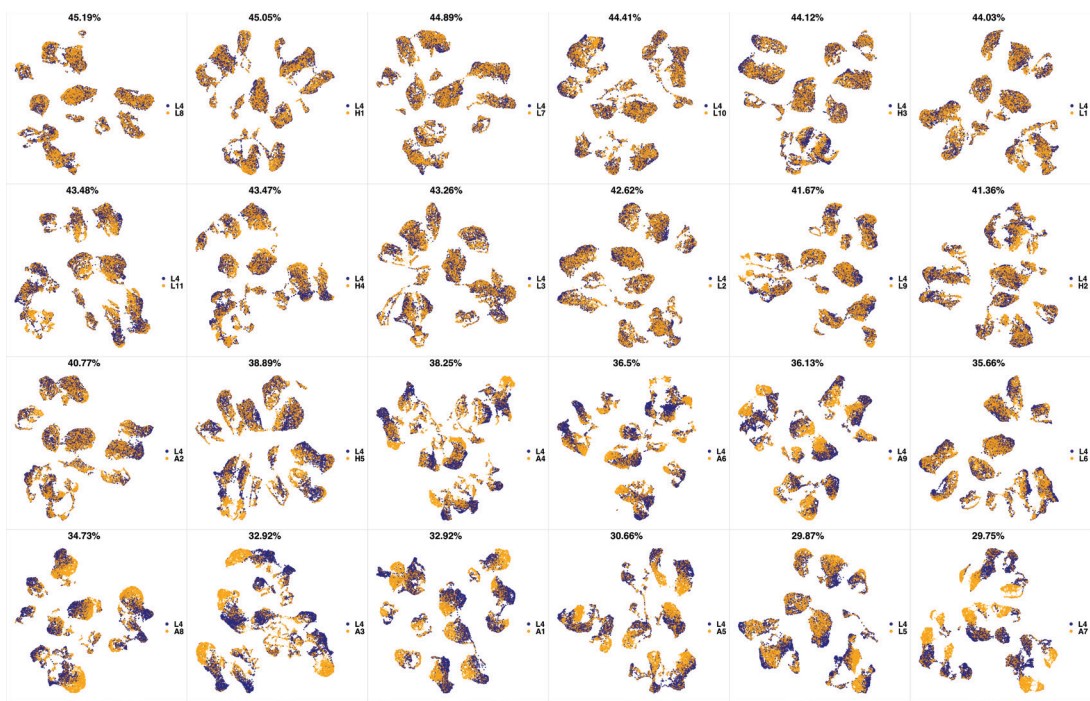

**Appendix 1—figure 19.** UMAP projections of L4 sample against all other patient samples. From top left to bottom right, the similarity measured by compaRe decreases as the degree of overlap decreases and the number of exclusive cell populations increases.

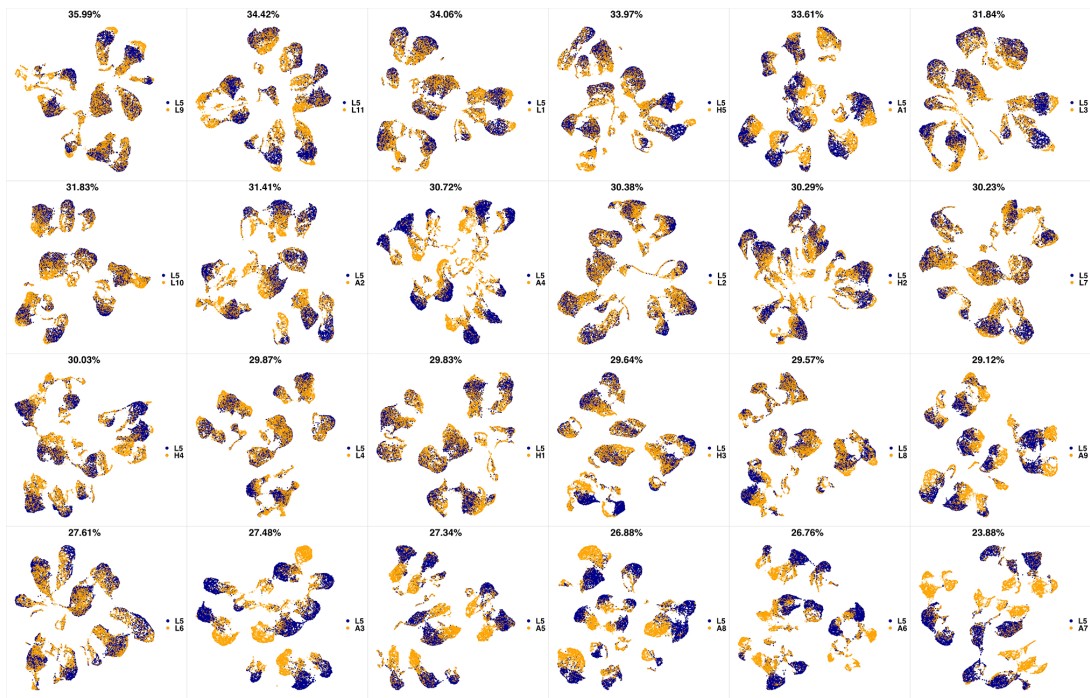

**Appendix 1—figure 20.** UMAP projections of L5 sample against all other patient samples. From top left to bottom right, the similarity measured by compaRe decreases as the degree of overlap decreases and the number of exclusive cell populations increases.

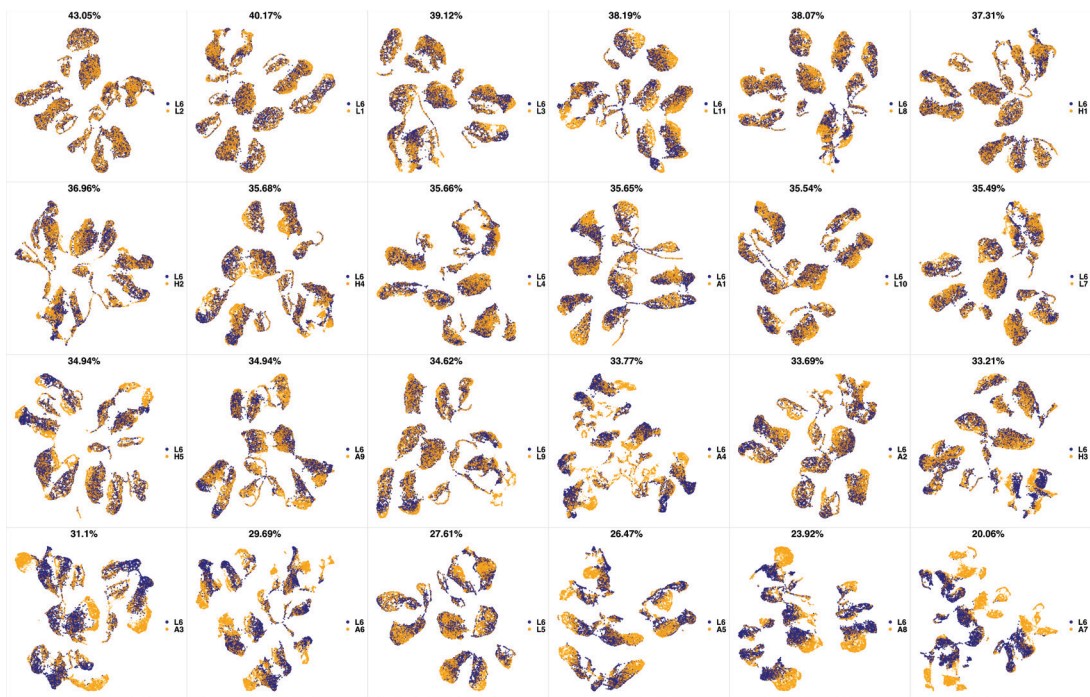

**Appendix 1—figure 21.** UMAP projections of L6 sample against all other patient samples. From top left to bottom right, the similarity measured by compaRe decreases as the degree of overlap decreases and the number of exclusive cell populations increases.

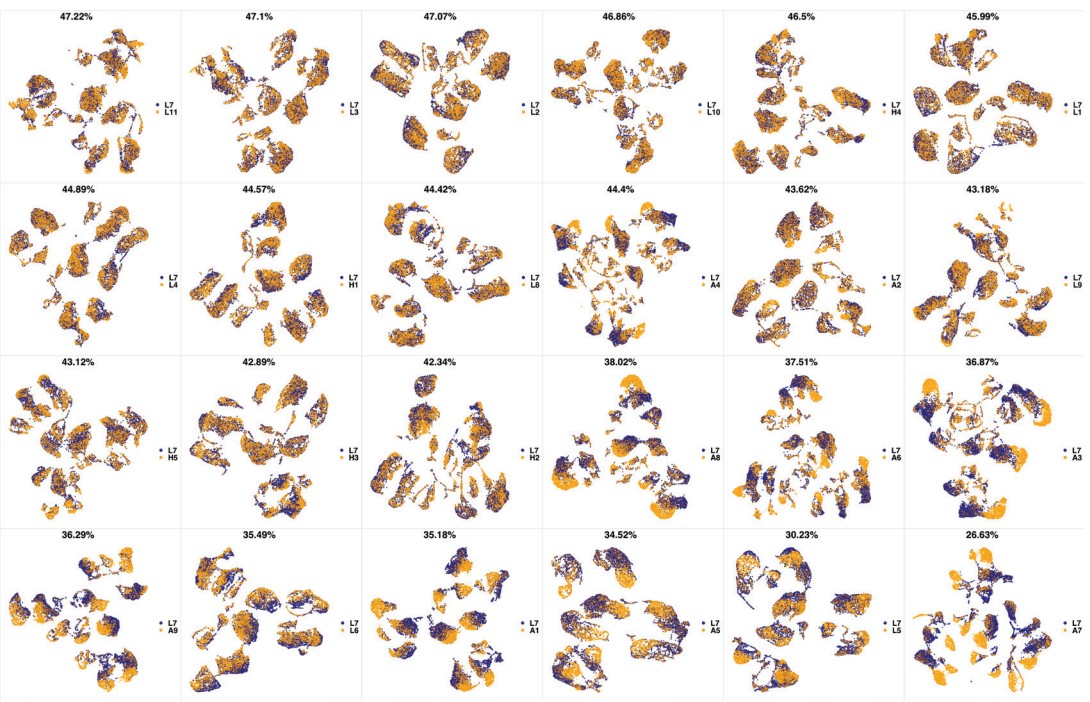

**Appendix 1—figure 22.** UMAP projections of L7 sample against all other patient samples. From top left to bottom right, the similarity measured by compaRe decreases as the degree of overlap decreases and the number of exclusive cell populations increases.

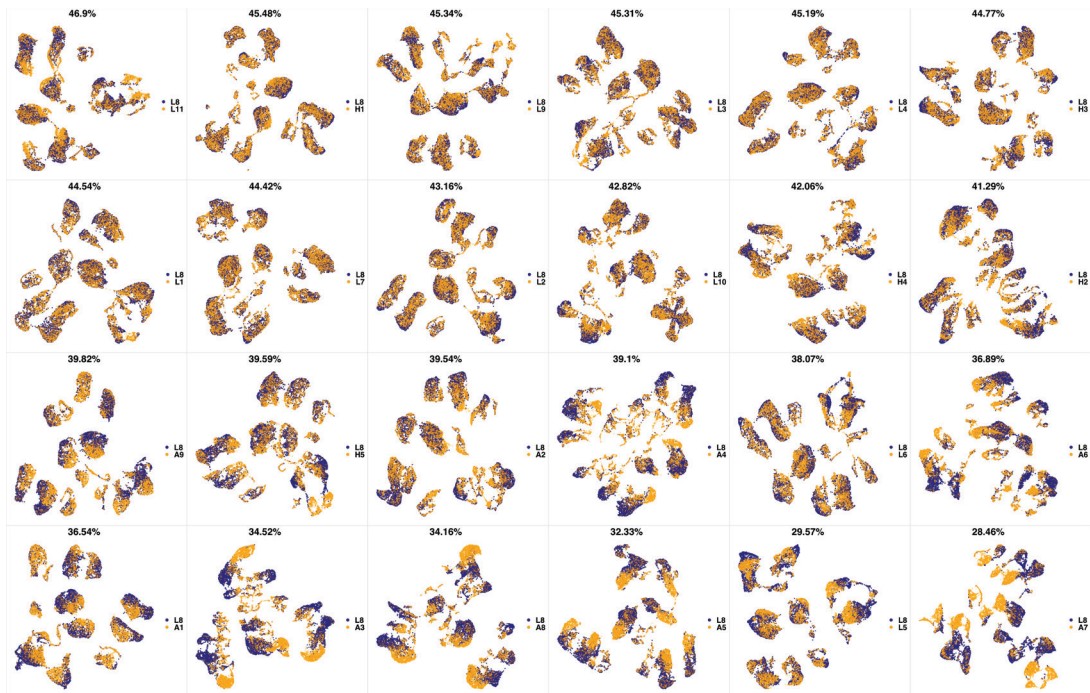

**Appendix 1—figure 23.** UMAP projections of L8 sample against all other patient samples. From top left to bottom right, the similarity measured by compaRe decreases as the degree of overlap decreases and the number of exclusive cell populations increases.

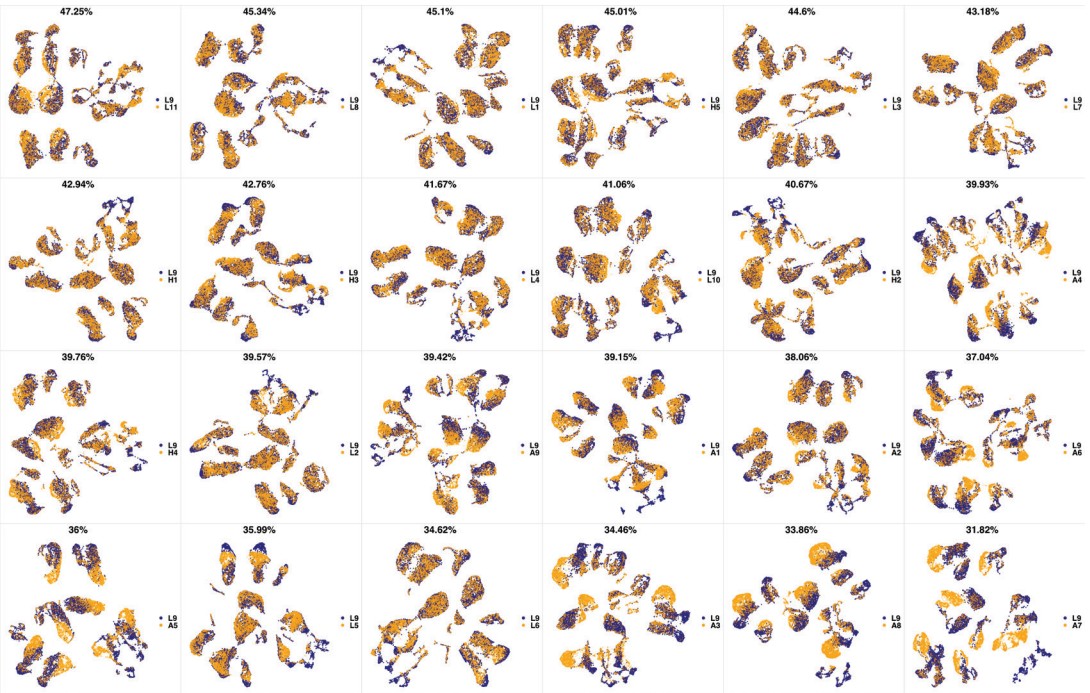

**Appendix 1—figure 24.** UMAP projections of L9 sample against all other patient samples. From top left to bottom right, the similarity measured by compaRe decreases as the degree of overlap decreases and the number of exclusive cell populations increases.

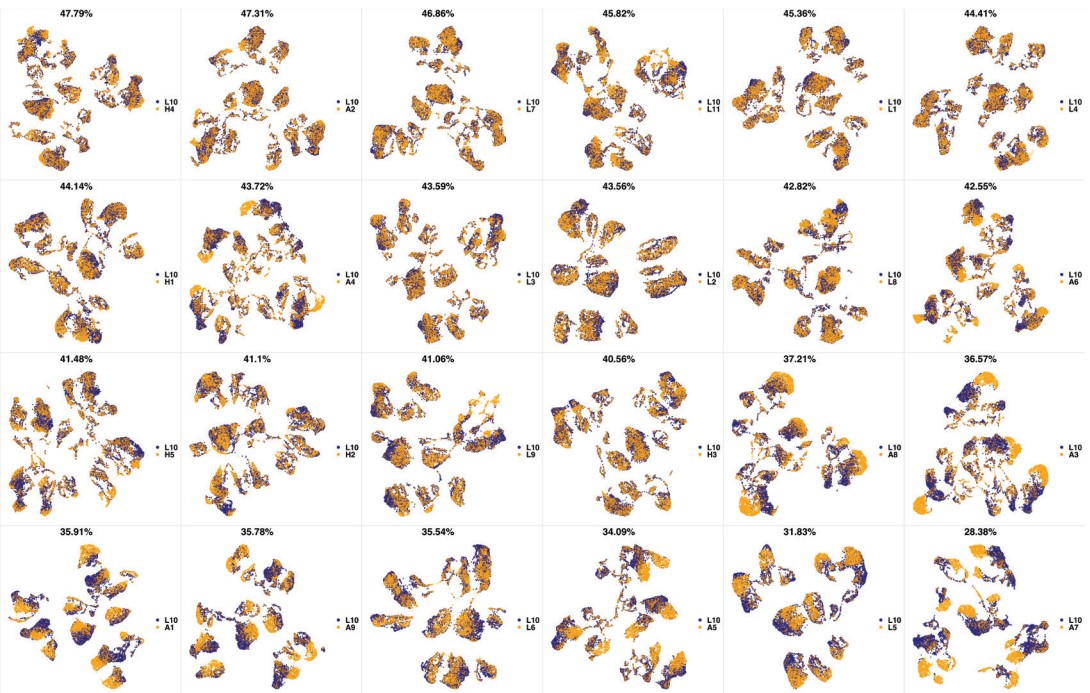

**Appendix 1—figure 25.** UMAP projections of L10 sample against all other patient samples. From top left to bottom right, the similarity measured by compaRe decreases as the degree of overlap decreases and the number of exclusive cell populations increases.

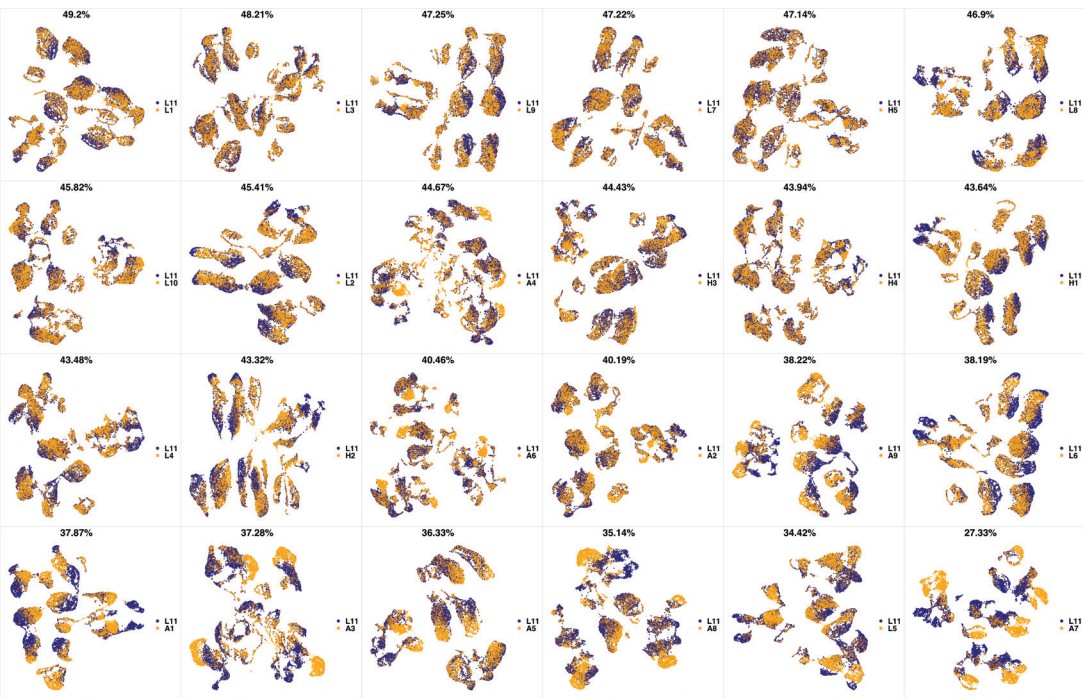

**Appendix 1—figure 26.** UMAP projections of L11 sample against all other patient samples. From top left to bottom right, the similarity measured by compaRe decreases as the degree of overlap decreases and the number of exclusive cell populations increases.

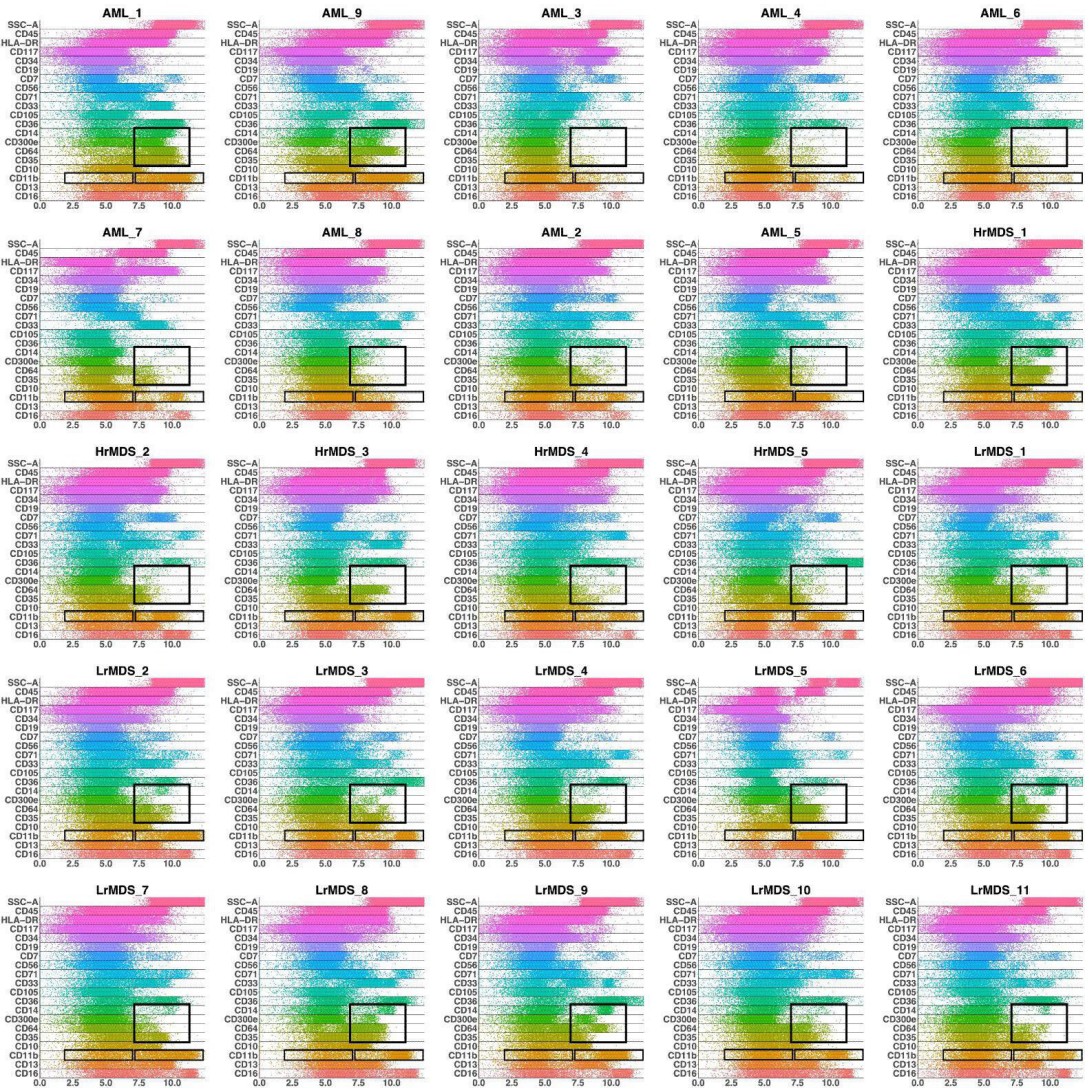

**Appendix 1—figure 27.** Band plots of AML and MDS patient samples. The immunophenotype of each patient sample is shown in a multiparameter band-dot plot (HrMDS: higher-MDS, LrMDS: lower-MDS). Rectangles gate positive and/or negative populations of monocytic maturation markers as well as the CD11b marker.

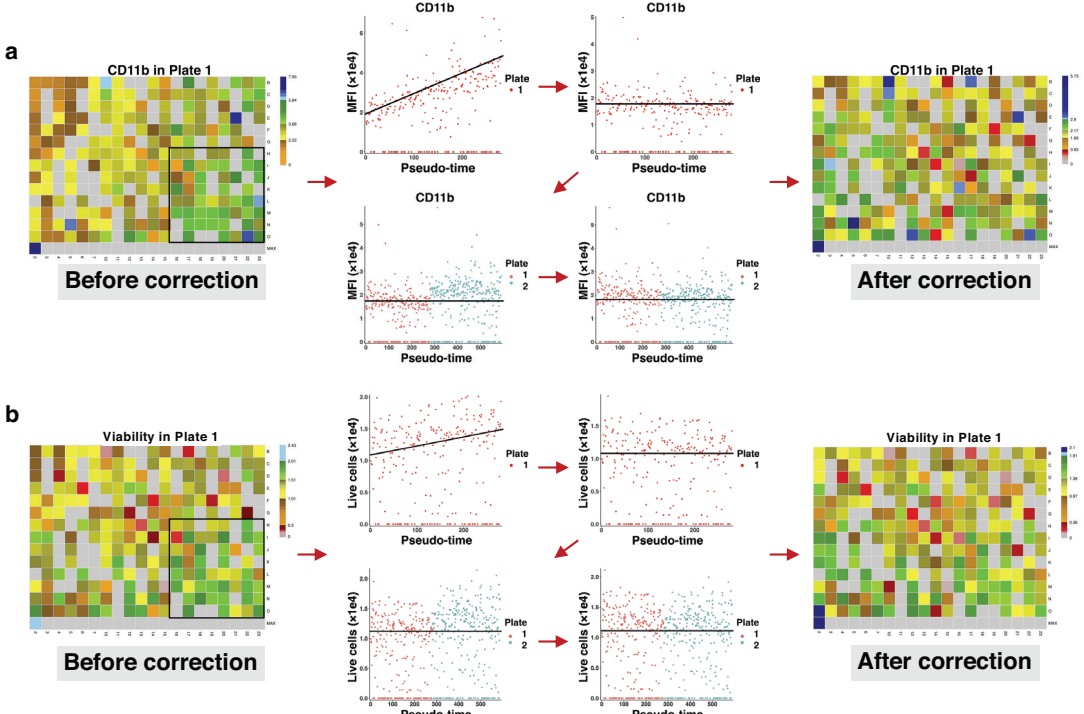

**Appendix 1—figure 28.** Correcting signal and cell viability drift. (**a**) Intra- and inter-plate signal drift correction. Accumulation of green-blue tiles in the bottom right corner of the left heatmap shows signal drift in CD11b expression for drugs in plate 1. Sorting median expressions (MFIs) of wells into reading order (column wise left to right) reveals a linear slope. After correction, the slope becomes non-positive (intra-plate correction). Still, there are different baselines between the two plates. Matching median lines of corrected values of all plates correct for this bias (inter-plate correction). (**b**) Intra- and inter-plate cell viability correction. Accumulation of green tiles in the bottom right corner of the left heatmap shows cell viability drift (7-AAD marker). We follow similar steps with (**a**) for cell viability correction.

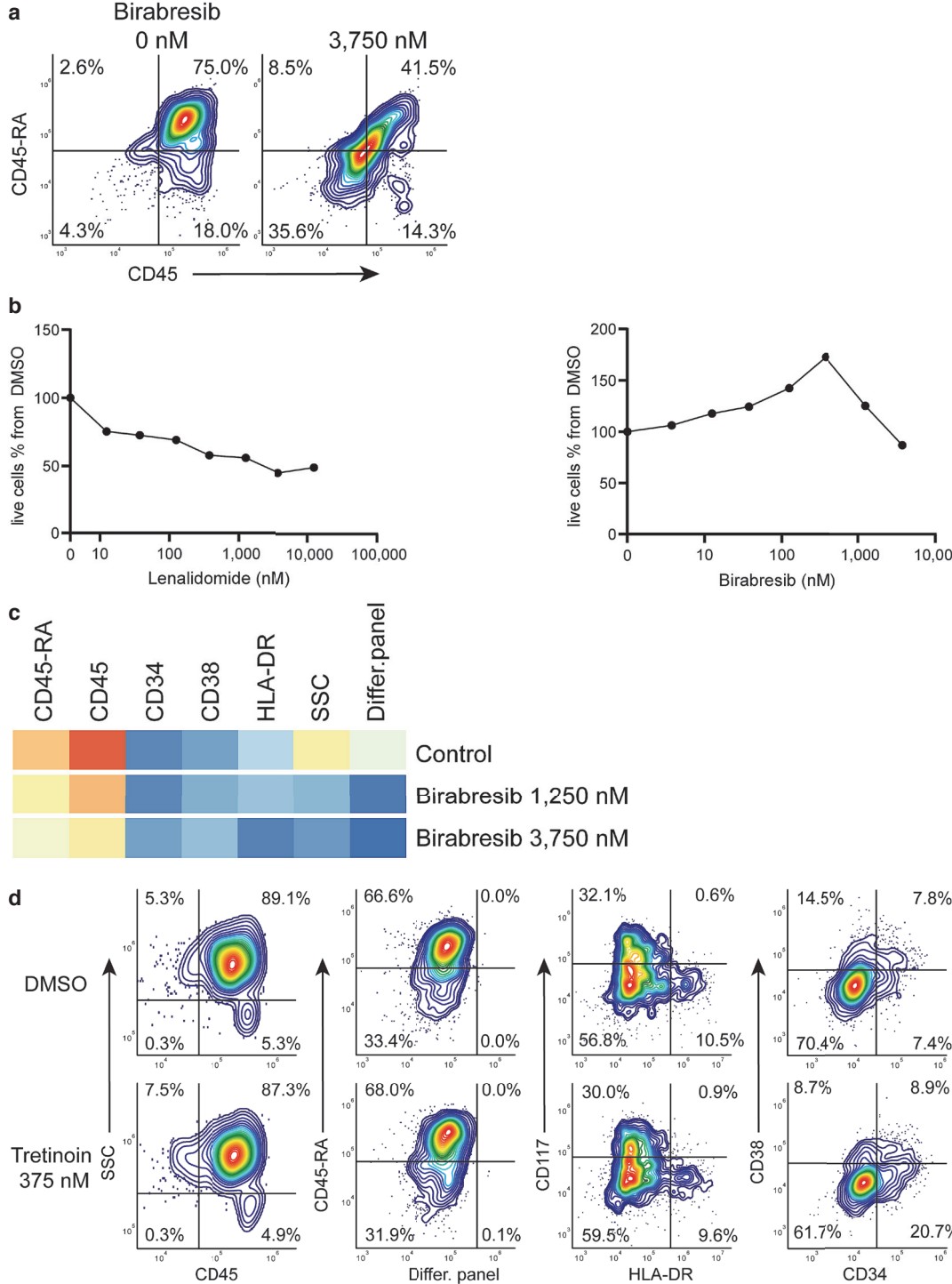

**Appendix 1—figure 29.** Birabresib treatment leads to loss of CD45 and CD45-RA expression without loss of live cell numbers. (**a**) Birabresib response as density scatter plot, CD45 vs CD45-RA. (**b**) Count of live cells per different concentrations of lenalidomide and birabresib. (**c**) Heatmap of birabresib response in all marker channels. (**d**) Example of response group 3: density scatter plots of DMSO-control vs. tretinoin 375 nM in different marker channels.

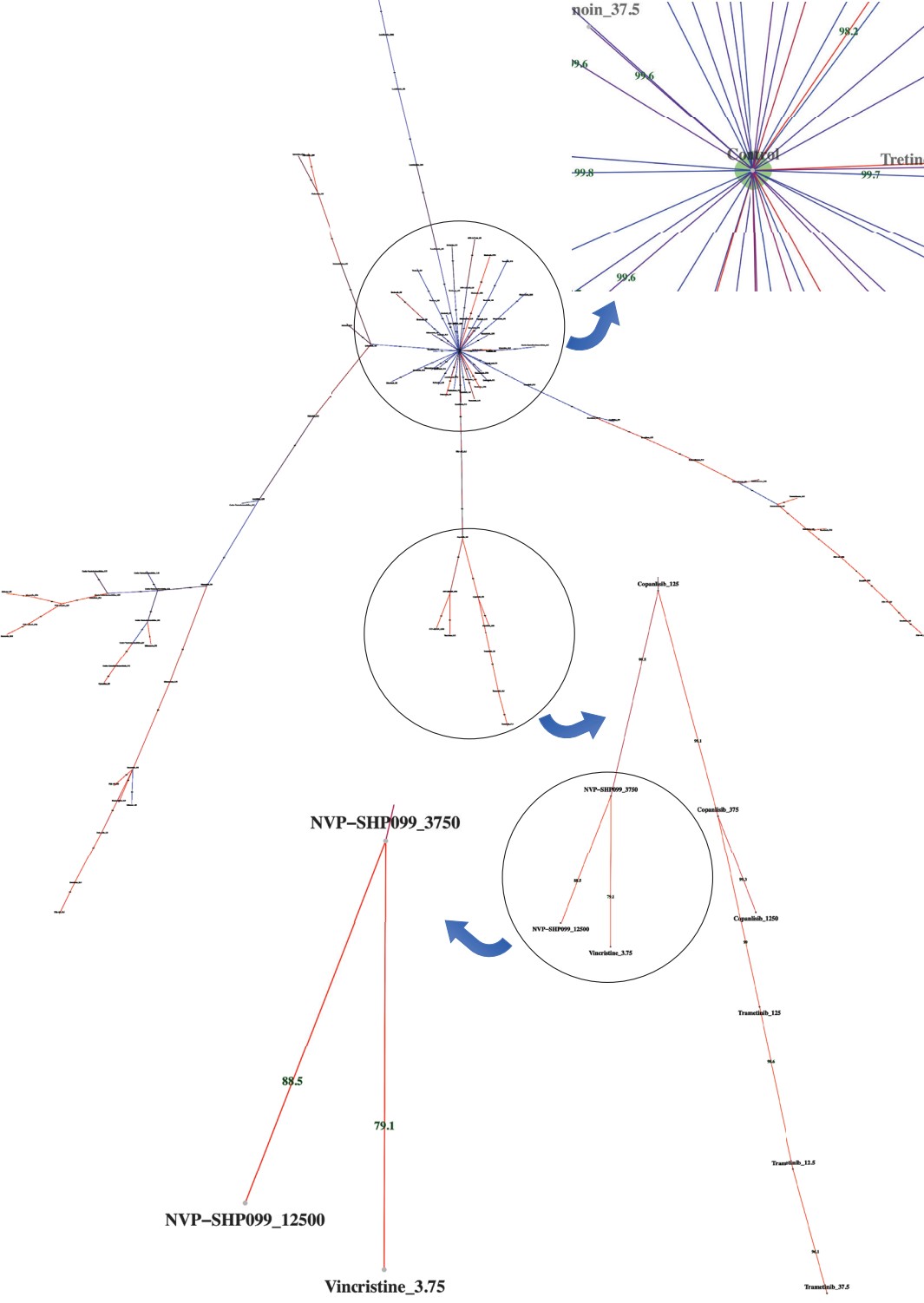

**Appendix 1—figure 30.** Dispersion graph. The (maximum spanning) tree demonstrates the dispersion of tens of potent antineoplastic agents around the control node containing negative controls (DMSO) and impotent agents. The drug library was analyzed by high-throughput flow cytometry coupled with compaRe in an AML human sample. Edge color and label show the amount of similarity between the agents. Impotent drugs are those which were similar enough to negative controls for a cutoff inferred during clustering. As the tree branches and spreads, drugs with stronger potency, usually with higher doses, tend to lie farther from the control node. Using the graph, the investigator can easily pick potent agents such as hits. The graph may also be potentially used to investigate different paths for mechanism of action, leading to different branches.

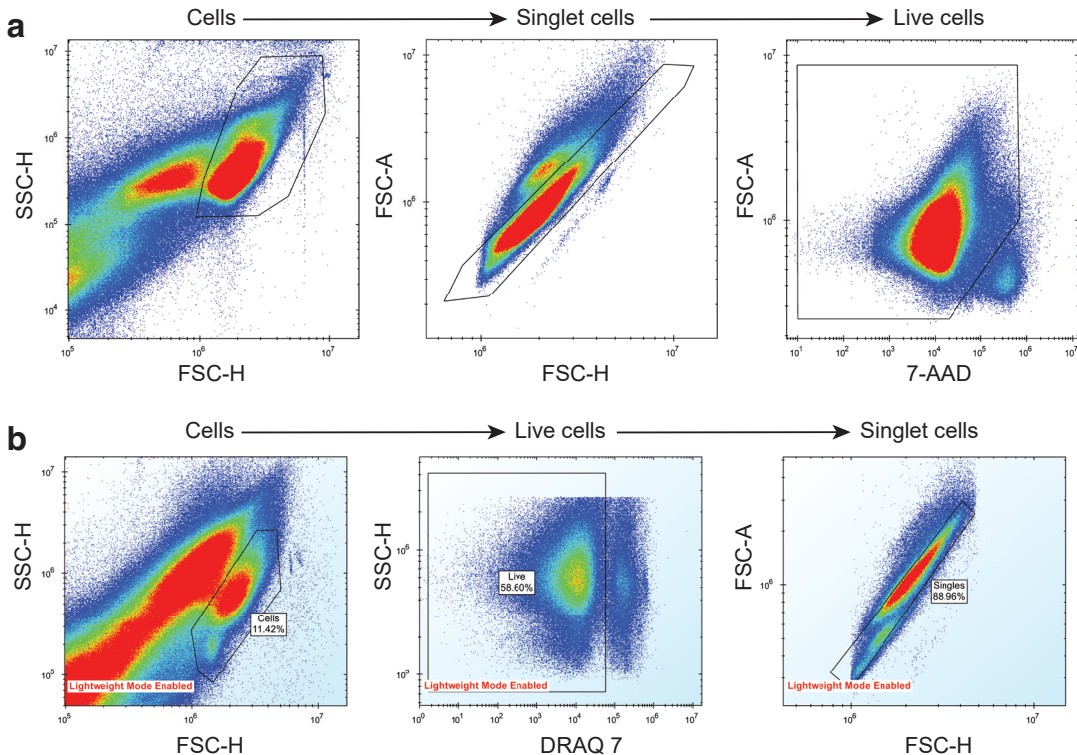

**Appendix 1—figure 31.** Removal of noise, dead cells and doublet cells from mouse and human AML sample drug screening data. (**a**) AML mouse model drug screening. (**b**) AML human sample drug screening. Cells were separated from debris using a side scatter height (SSC-H) vs forward scatter height (FSC-H) plot. Singlet cells were determined from FSC-H vs forward scatter area (FSC-A) plot. Live cells were separated from dead cells using a dead-cell-labelling dye, either 7-AAD or DRAQ7.

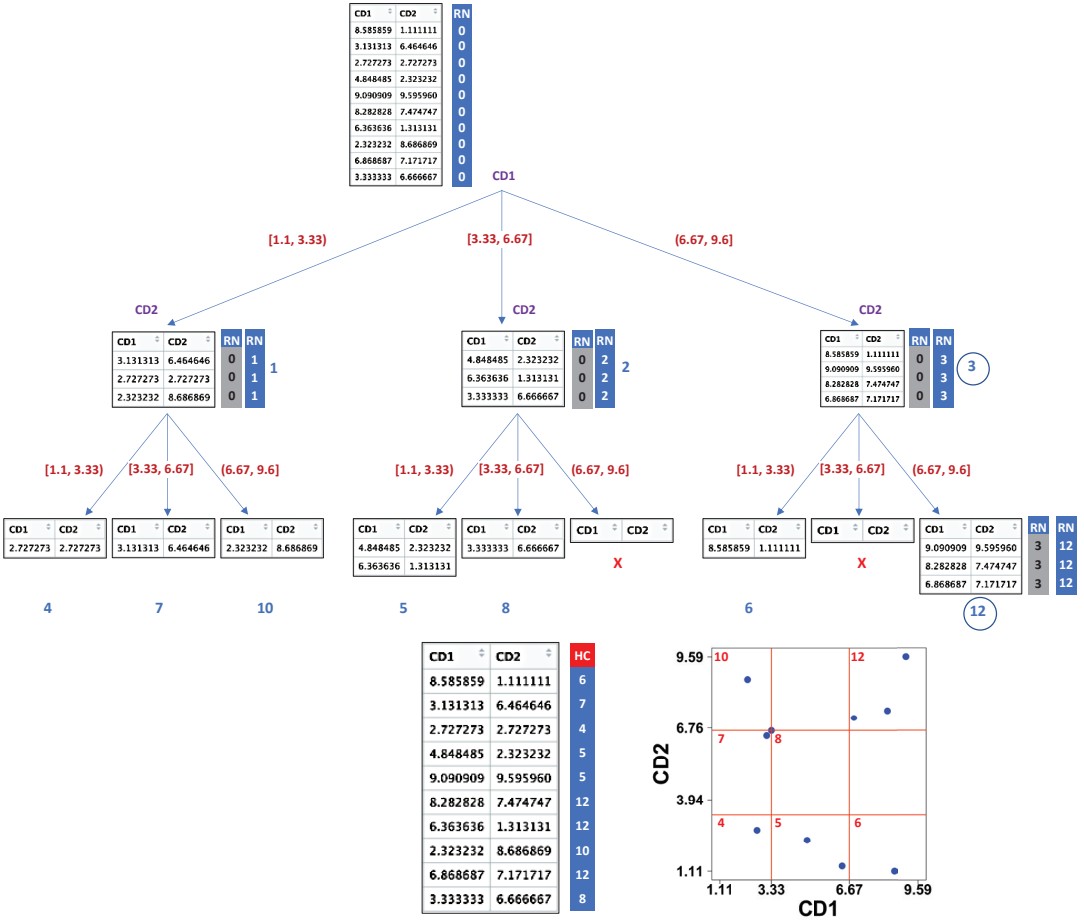

**Appendix 1—figure 32.** Demonstration of compaRe algorithm using a 2-dimensional table. It first forms an abstract square grid (red) encompassing all the data points within the range (1.1, 9.6). At the top level, all the cells (table rows) are in the region number (RN) 0. First iteration divides the first dimension formed by CD1 marker into 3 ( = *n*) subsets. Assuming a left-first numbering rule, the RN column is dynamically updated (blue column) for each subset using some information such as current RN (grey column), current dimension and possible number of families and siblings behind. For instance, child node 12 has parent node 3, could have two siblings (node 6, node 9) and two families (parent 1, parent 2) behind, although children 11 and 9 were never born as marked with X. Final leaves are called hypercubes (HCs). The corresponding grid on the biplot demonstrates that two regions which were devoid of data points have not been assigned any hypercube. For comparing two samples, they are first jointly normalized between a range. The tree graph is just for better visualization and will not be implemented.

