## [Editor Report]

This paper aims to address the current gap in the efficient analysis of large-scale multiparameter flow cytometry and other datasets. The authors offer a software toolkit with an efficient algorithm for comparing numerous samples at once. The study is well presented and is relevant to single cell analysis research.

---

## [Decision Letter]

**Decision letter after peer review:**

Thank you for submitting your article "Comprehensive and unbiased multiparameter high-throughput screening by compaRe finds effective and subtle drug responses in AML models" for consideration by *eLife*. Your article has been reviewed by 3 peer reviewers, and the evaluation has been overseen by a Reviewing Editor and Aleksandra Walczak as the Senior Editor. The following individual involved in review of your submission has agreed to reveal their identity: Aik Choon Tan (Reviewer #2).

Essential revisions:

Please address all the reviewers comments. The reviewers agree it is a strong paper that is well written but certain improvements can still be made. Especially please give the reader more context in the introduction and compare your method with other existing methods.

*Reviewer #1:*

Comprehensive and unbiased multiparameter high-throughput screening by compaRe finds effective and subtle drug responses in AML models by Hajkariim et al. introduces a pipeline for pre-processing and analyzing data from multiplex flow cytometry and other technologies. Preprocessing steps include algorithms for correcting common sources of bias in such data. Another key feature is a robust approach to measuring cell similarity across samples. Among the strengths are that the manuscript is well-written, the analysis pipeline is well-motivated, and illustrated with apt examples. The similarity measure is very interesting as well.

There are a few weaknesses as well. It is not completely clear to me how this pipeline agrees and disagrees with common practice in the field. References 1-3, cited to document ongoing analytic challenges, are all at least 5 years old. Comparisons to other approaches, including the use Jensen-Shannon Divergence for similarity, make a convincing case that the proposed method is both effective and computationally efficient, but it is not clear if the comparators represent true standard of practice, or mere straw men. Methodologies are complex and can be difficult to follow, especially the similarity measure.

I would like to see the current state of the field described more clearly in the introduction, as context for the current effort. What makes this unique and important today? This type of clustering is common in single cell sequencing analysis, is it also commonly done in multiplex flow etc, maybe with less computationally demanding tools than JSD? If not, why not?

I think I understand how the similarity measure here works, though its hard to follow the details. Its very interesting, but to be honest, I can't decide if I think its a good solution or needlessly complex. The key, as I understand it, is the binning into "relative" expression groups. This is, after all, how 2-dimensional flow data is commonly interpreted – with plots treated as a visual 2 X 2 table, with row and column boundaries existing largely in the eye of the beholder. The methods need to be clearer throughout, esp. this part. It might help to add another author, a third party who has to understand the method from scratch, and who, having not lived with the details as long, might be expected to provide a more user friendly sense of the overall approach.

*Reviewer #2:*

In this manuscript, Hajkarim et al. developed compaRe, a user friendly software suite (written in R) for analyzing high-throughput, multi-parameter screening data. There are several modules included in the compaRe toolkit, which can be individually invoked to perform specific tasks, such as quality control, bias correction, pairwise comparisons, clustering and data visualization. All of these modules are available as command-line version and a GUI version for users to use in data analysis, visualization and results interpretation. The authors showed the utility of their toolkit in analyzing multiparameter mass and flow cytometric data from AML and MDS patient samples. Through this analysis using compaRe, the authors showed that they can identify patient heterogeneity and drug response profiles. Overall, this is a well organized and written manuscript describing the development of the new compaRe toolkit. The method is clearly described, and the user manual/tutorial is easy to follow. It seems like compaRe will be a useful toolkit for the research community, which is eager for a one-stop pipeline for analyzing high-throughout multiparameter screening data.

et al.

Strengths:

1. All of these modules are available as command-line version and a GUI version for users to use in data analysis, visualization and results interpretation.

2. The authors showed the utility of their toolkit in analyzing multiparameter mass and flow cytometric data from AML and MDS patient samples. Through this analysis using compaRe, the authors showed that they can identify patient heterogeneity and drug response profiles.

3. Overall, this is a well organized and written manuscript describing the development of the new compaRe toolkit. The method is clearly described, and the user manual/tutorial is easy to follow.

4. It seems like compaRe will be a useful toolkit for the research community, which is eager for a one-stop pipeline for analyzing high-throughout multiparameter screening data.

Weaknesses:

1. The current manuscript didn't compare with some other existing programs/software in analyzing flow and mass cytometry data. It will be important to compare compaRe with existing tools, to show the strengths and weaknesses of compaRe with other tools.

2. The authors could think about adding an additional module to integrate other "omics" data (e.g. such as mutational or gene expression/signatures or pathways), this could be useful for doing the clustering step or to identify patients having the same mutational profiles.

*Reviewer #3:*

Hajkarim et al. implement an algorithm in their presented toolkit compaRe to compare samples based on the similarities of samples, distinct from the more commonly used meta-clustering approaches, such as PhenoGraph, or dimensional reduction with Jenssen-Shannon Divergence analysis. Similarities among samples are calculated based on the proportions of cells within a sample belonging to an n-dimensional "hypercubes" (or "hypergridding" that is actually mass-aware and not blind) that are stratified by expression levels for n number of markers. The authors demonstrate that this method is much more time-efficient, obviates subsampling, and is robust to batch effects. This method is particularly appropriate for large-scale datasets, facilitating the comparison of numerous samples which would be helpful in screening efforts. The manuscript is written and presented well.

Major strengths:

1. The study demonstrates sufficiently strong support for the toolkit's ability to determine similarity across samples and its computing efficiency with Figure 2, an important advantage of this tool.

2. Compared to other approaches, the method is advantageous for identifying groups of samples that may be similar in a very large-scale dataset. CompaRe does not require (or make use of) manual expert annotation of meta-clusters. The workflow is efficient and unbiased.

Major weakness:

A major weakness of the current presentation of the study is that it has not clearly demonstrated the toolkit's utility in exploring specific phenotypes in-depth within a high-parameter dataset. The following are two examples in which this limitation is relevant, and the authors may address this to strengthen this paper, if in fact detailed phenotyping is considered by the authors as an important feature of the toolkit. If not, the authors should revise the manuscript as such.

First, the authors stated that their approach can be used to "optimize true cytometric n-dimensional immunophenotypic characterization" even in the setting of multi-panel workflow. However, the demonstration was based on samples that seem to have predominant phenotypes that are almost mutually exclusive. It is unlikely that this toolkit would be useful for reliable phenotypic characterization in a largely heterogeneous population of cell types, e.g. even in normal peripheral blood, unless a high number of parameters was concurrently acquired within the dataset. This is an inherent limitation. Nonetheless, a revision to demonstrate how compaRe can evaluate specific clusters of phenotypes with biological significance from a high-parameter dataset (20-30 marker cytometry) would be very helpful.

Second, the authors refer to the method's ability to include all rare cell subsets in the analysis, i.e. the ability to forego any subsampling. The work does not, however, demonstrate clearly how the presence of a rare cell subset in a given sample influences its similarity to other samples. Thus, the toolkit's value of being able to include such a rare subset in the analysis remains unestablished. It would be beneficial to include such a test to see whether the algorithm is sensitive to such changes.

---

## [Author Response]

Essential revisions:Please address all the reviewers’ comments. The reviewers agree it is a strong paper that is well written but certain improvements can still be made. Especially please give the reader more context in the introduction and compare your method with other existing methods.Reviewer #1:[…] There are a few weaknesses as well. It is not completely clear to me how this pipeline agrees and disagrees with common practice in the field. References 1-3, cited to document ongoing analytic challenges, are all at least 5 years old. Comparisons to other approaches, including the use Jensen-Shannon Divergence for similarity, make a convincing case that the proposed method is both effective and computationally efficient, but it is not clear if the comparators represent true standard of practice, or mere straw men. Methodologies are complex and can be difficult to follow, especially the similarity measure.I would like to see the current state of the field described more clearly in the introduction, as context for the current effort. What makes this unique and important today? This type of clustering is common in single cell sequencing analysis, is it also commonly done in multiplex flow etc, maybe with less computationally demanding tools than JSD? If not, why not?

Thank you for your comments. We believe compaRe is a unique tool in that it is designed specifically to cluster samples. Other methods that are commonly used for the analysis of mass/flow cytometry data are primarily designed to cluster cells within samples. The obtained cell clustering information is subsequently used for assessing similarity between samples by meta-clustering. Also, using clustering-free algorithms is often computationally expensive and not as robust as one may wish. For example, JSD was employed to compare samples, but it is not efficient for comparing a large number of samples, and, as we showed in the results, is computationally expensive.

We had previously highlighted these limitations by comparing the outcome of a compaRe analysis to meta-clustering and JSD. We have now provided the comparisons with other well-established and commonly used cell clustering approaches for mass/flow cytometry: FlowSOM and SPADE (main text lines 113-115 and Appendix 1-figure 31). To make these points clearer, we have modified the Introduction. Specifically, we added that:

i. Main text lines 33-35: ‘These technologies can test hundreds of samples (such as drug treatments) each with tens of thousands of events (e.g., cells) labeled for numerous biomarkers (such as cytoplasmic or membrane markers).’

ii. Main text lines 43-48: ‘Meta-clustering with single-cell clustering algorithms has been suggested to cluster samples based on the similarity of the centroids of cell subpopulations identified in the individual samples (5-8). While these algorithms are widely used in single-cell data analysis for clustering cells, they are not efficient for clustering of samples. This is because centroid-based analysis can be misleading when subclusters are not sufficiently distinct or the number of sub-clusters varies.’

iii. Main text 56-58: ‘Available computational toolkits mostly allow for single-parameter or unautomated analyses of large-scale screening data using the aforementioned methods. In these toolkits, each well should be first represented by a single parameter such as cell counts or centroids or they require manual intervention.’

I think I understand how the similarity measure here works, though it’s hard to follow the details. It’s very interesting, but to be honest, I can't decide if I think it’s a good solution or needlessly complex. The key, as I understand it, is the binning into "relative" expression groups. This is, after all, how 2-dimensional flow data is commonly interpreted – with plots treated as a visual 2 X 2 table, with row and column boundaries existing largely in the eye of the beholder. The methods need to be clearer throughout, esp. this part. It might help to add another author, a third party who has to understand the method from scratch, and who, having not lived with the details as long, might be expected to provide a more user friendly sense of the overall approach.

Dynamic programming is key for reducing processing power. We try to explain the concept more clearly as follows (also added to the Appendix 1 lines 114-120). As you pointed out, the goal is to bin/grid data into relative expression groups (hypercubes). Gridding can be implemented by a simple algorithm dividing each dimension in each iteration. However, after a couple of rounds, this naïve algorithm turns out to be infeasible. This can be shown through visualizing the gridding algorithm by a tree graph (Appendix 1-figure 29). The tree clearly shows that the comparison complexity of the gridding algorithm is exponential since the tree grows exponentially. Therefore, we were forced to use a more efficient algorithm for implementing gridding. Dynamic programming (algorithm) turned out to be quite effective. What makes dynamic programming very effective is its ability to memorize the values computed in the previous iterations avoiding recomputing potentially expensive algebraic operations (Appendix 1-equation 3). Moreover, considering that any signal drift correction is essentially an approximate method, the robustness of this algorithm to noise creates a synergistic effect. Therefore, we believe that the current description about similarity measure is required.

Reviewer #2:[…] 1. The current manuscript didn't compare with some other existing programs/software in analyzing flow and mass cytometry data. It will be important to compare compaRe with existing tools, to show the strengths and weaknesses of compaRe with other tools.

Thank you for your comments. We previously benchmarked compaRe’s similarity module against meta-clustering with several cell clustering methods and selected widely used PhenoGraph which performed better than others (considering several factors). We also used JSD as a clustering-free approach. In response to the request from the reviewer, we have now added the comparisons with SPADE and FlowSOM, two other commonly used algorithms for the analysis of flow and mass cytometry data (main text lines 113-115 and Appendix 1-figure 31). However, please note that available automated and commonly used tools are primarily cell clustering/automatic gating approaches. These include methods such as PhenoGraph, SPADE, and FlowSOM. A key reason why compaRe outperforms these and other methods is that they are designed to cluster cells rather than samples, while compaRe clusters samples.

2. The authors could think about adding an additional module to integrate other "omics" data (e.g. such as mutational or gene expression/signatures or pathways), this could be useful for doing the clustering step or to identify patients having the same mutational profiles.

Drug response in association with genetic alterations is one of the applications of compaRe. The genetic alteration can be visualized in the clusters that compaRe identifies. However, at the moment, we do not have relevant data to apply this type of analysis for visualization. To highlight that the functionality for this type of analyses exists within compRe, we have now explained this in the main text lines 212-213.

Reviewer #3:[…] A major weakness of the current presentation of the study is that it has not clearly demonstrated the toolkit's utility in exploring specific phenotypes in-depth within a high-parameter dataset. The following are two examples in which this limitation is relevant, and the authors may address this to strengthen this paper, if in fact detailed phenotyping is considered by the authors as an important feature of the toolkit. If not, the authors should revise the manuscript as such.First, the authors stated that their approach can be used to "optimize true cytometric n-dimensional immunophenotypic characterization" even in the setting of multi-panel worklow. However, the demonstration was based on samples that seem to have predominant phenotypes that are almost mutually exclusive. It is unlikely that this toolkit would be useful for reliable phenotypic characterization in a largely heterogeneous population of cell types, e.g. even in normal peripheral blood, unless a high number of parameters was concurrently acquired within the dataset. This is an inherent limitation. Nonetheless, a revision to demonstrate how compaRe can evaluate specific clusters of phenotypes with biological significance from a high-parameter dataset (20-30 marker cytometry) would be very helpful.

Thank you for your comments. To address this comment, we generated Appendix 1-figure 32 and added the result to the main text lines 124-131. As compaRe measures the similarity between cluster morphologies, we carried out an analysis in which cells from a cluster (an immunophenotypic cell population) were gradually removed to contort its configuration. We used a dataset of 3 healthy and 2 pediatric AML bone marrow mononuclear cell samples from the data provided in the 6th reference. Samples were stained with 29 (15 membrane and 14 intracellular signaling) markers. Taking H1 as reference, we gradually removed 25%, 50%, 75% and 100% (phenotypic changes) of cells from a target cluster identified by PhenoGraph. As the UMAP projections show, the similarity decreased concurrently and more drastically when phenotypic changes were detected.

Second, the authors refer to the method's ability to include all rare cell subsets in the analysis, i.e. the ability to forego any subsampling. The work does not, however, demonstrate clearly how the presence of a rare cell subset in a given sample influences its similarity to other samples. Thus, the toolkit's value of being able to include such a rare subset in the analysis remains unestablished. It would be beneficial to include such a test to see whether the algorithm is sensitive to such changes.

We previously discussed in the manuscript that ‘the poor performance of JSD indicates that this approach can work well only in the absence of signal shift. It is of particular note that compaRe does not need subsampling or dimension reduction of the input data, avoiding the risk of losing rare subpopulations.’ The use of full data rather than subsampled data allows for more robust analyses, in part, simply, owing to using more data and the ability to establish a more refined overall cell cluster morphology of each sample.

Additionally, in the analysis explained above (Appendix 1-figure 32), the 75% sample potentially resembles a rare cell population, a cell cluster with a small number of cells. The analysis demonstrated compaRe was sensitive enough to maintain a certain similarity in the presence of a rare cell population, as the similarity score was reduced when this population was gone in the 100% sample. Nonetheless, since subsampling does not necessarily lose rare subpopulations and to avoid confusion, we removed the phrase ‘avoiding the risk of losing rare subpopulations’ from the discussion.